# Hot carriers perspective on the nature of traps in perovskites

Marcello Righetto [1,3], Swee Sien Lim[1,3], David Giovanni [1,3], Jia Wei Melvin Lim [1,2], Qiannan Zhang [1], Sankaran Ramesh [1,2], Yong Kang Eugene Tay [1] & Tze Chien Sum [1✉]

Amongst the many spectacular properties of hybrid lead halide perovskites, their defect tolerance is regarded as the key enabler for a spectrum of high-performance optoelectronic devices that propel perovskites to prominence. However, the plateauing efficiency enhancement of perovskite devices calls into question the extent of this defect tolerance in perovskite systems; an opportunity for perovskite nanocrystals to fill. Through optical spectroscopy and phenomenological modeling based on the Marcus theory of charge transfer, we uncover the detrimental effect of hot carriers trapping in methylammonium lead iodide and bromide nanocrystals. Higher excess energies induce faster carrier trapping rates, ascribed to interactions with shallow traps and ligands, turning these into potent defects. Passivating these traps with the introduction of phosphine oxide ligands can help mitigate hot carrier trapping. Importantly, our findings extend beyond photovoltaics and are relevant for low threshold lasers, light-emitting devices and multi-exciton generation devices.

[1] Division of Physics and Applied Physics, School of Physical and Mathematical Sciences, Nanyang Technological University, 21 Nanyang Link, Singapore 637371, Singapore. [2] Energy Research Institute @NTU (ERI@N), Interdisciplinary Graduate School, Nanyang Technological University, 50 Nanyang Avenue, Singapore 639798, Singapore. [3] These authors contributed equally: Marcello Righetto, Swee Sien Lim, David Giovanni. ✉email: Tzechien@ntu.edu.sg

Hybrid lead halide perovskites (LHPs) are thrust into the limelight of semiconductor research by the advances they bring into the field of solution-processed optoelectronics[1] spanning photovoltaics[2], light-emitting devices[3], lasers[4], and spintronics[5]. Surprisingly, LHP-based devices can achieve extraordinary performances despite possessing large densities of point defects. These defects arise from solution-processing (about $10^{16}$ to $10^{17}$ cm$^{-3}$ for thin films) and consist predominantly of shallow traps with a smaller population of deep trap sites[6–8]. The latter act as non-radiative Shockley-Read-Hall recombination centers and are detrimental for device operations[9,10]. Past defect studies focused almost exclusively on deep trap states (such as lead vacancies, $V_{Pb}$), trying to mitigate their detrimental role in LHP by formulating design and synthetic rules[11–13]. In contrast, carriers trapped in shallow defect sites have a high probability of de-trapping and are thus believed not to contribute significantly to non-radiative recombination rates[8]. Because these shallow traps do not affect the electronic and optical properties of LHPs, this class of materials is commonly termed 'defect-tolerant'[14,15].

After several years of relentless growth, efficiency enhancements in perovskite-based devices are reaching a plateau. This calls for a more in-depth study of defects in LHPs, a reconsideration of the role of shallow traps and their bearing on defect tolerance[13]. Many of these applications rely on high-energy, non-resonant excitations, e.g., GaN back-illumination or high-energy charge injection (around 1 eV excess energy above the band-gap) in light-emitting devices, and violet/blue sunlight conversion in solar cells. Hence, for them to achieve high performances, the defect tolerance of LHPs should also extend to high-energy excitations. The study of excitation energy-resolved photo-luminescence quantum yield (PLQY) provides an insight into how incident energy affects the interaction between photo-generated carriers and traps. For instance, in molecular systems, the fast internal conversion process (i.e., cooling) and the absence of traps make the PLQY unsusceptible to the excitation energy[16,17]. It is essential to note that excitation energy-dependent PLQYs have a direct impact on device performance, and they can occur in all condensed matter systems due to the coupling between hot carriers and traps. Several recent works[18,19] reported conflicting results on the presence of this coupling, thereby leaving an open question.

Perovskite nanocrystals (PNCs) provide an exciting platform to investigate the extent of defect tolerance to higher energy states, owing to (i) their higher PLQY that facilitate the use of fluorescence techniques, (ii) the introduction of surfaces as inherently defective sites, and (iii) the possibility of controlling the passivation of these defects with effective surface chemistry tools[20–22]. In recent years, PNCs took the nanocrystals field by storm, outpacing traditional II-VI semiconductor NCs in many of their principal applications, e.g., lower MEG and ASE thresholds, longer-lived hot carriers and larger multiphoton cross-sections[23–26]. PNCs have narrow linewidths (12–30 nm), an ultra-wide color gamut (+140% of NTSC standard), and PLQYs reaching 90% and higher for bare core NCs[27,28]. The possibility of retaining high PLQY without extensive engineering of their surfaces, is the hallmark of a defect-tolerant electronic structure[27,29].

Herein, we demonstrate that MAPbX$_3$ (X = I, Br) PNCs exhibit a substantial excitation energy-dependence on the PLQY, thus setting a fundamental limit to their photon conversion efficiencies. Using pump-probe (PP) and pump-push-probe (PPP) ultrafast spectroscopy measurements supported with phenomenological modeling, we demonstrate how hot carrier trapping mechanisms cause these excitation energy-dependent carrier losses in PNCs. In addition, we prove that it is possible to reduce their traps' electronic coupling with hot carriers by performing a post-synthetic ligand exchange to passivate some of these traps.

By uncovering these mechanisms, we aim to provide deeper insights into the photophysics of defect-tolerant PNCs and show that this tolerance may not apply to hot carriers. Importantly, these findings exemplify the need for further efforts in developing synthetical approaches, such as post-synthetic treatments, alternative types of ligands, and core-shell heterostructures that can decouple and mitigate the deleterious effects of defects on the hot carriers in these systems.

## Results

**Excitation energy-dependent PLQY in PNCs.** We synthesized ambient-condition stable colloidal suspensions of MAPbX$_3$ (X = I, Br) nanocrystals by a modification of the previously reported ligand assisted reprecipitation (LARP) method (see "Methods")[30,31]. XRD demonstrates the formation of a cubic structure (space group $Pm\bar{3}m$), and TEM micrographs (Supplementary Figs. 1–3) confirm the quality of the synthesized samples: the particle size distribution obtained by statistical analysis is analogous to previous reports and between the two samples: MAPbBr$_3$ and MAPbI$_3$ NCs, with average radii of $4.2 \pm 1.3$ nm and $5.5 \pm 1.5$ nm, respectively. For these two samples, we studied the role of excess photoexcitation energy above the band-gap (henceforth referred to as $\delta_E$), in determining the PLQY of the PNCs. The effect of $\delta_E$ in the PL of molecular and inorganic solids reflects the efficiency of the internal conversion or intraband relaxation. The PLQY spectra plotted as a function of $\delta_E$, also known as 'photo-action' spectra, therefore contain information on hot carrier effects: dynamic processes occurring during the relaxation of high-energy carriers to the band edge[32]. These measurements were done with low-excitation intensities, as much as the techniques allow, to keep our focus on trapping and carrier loss mechanisms in the PNCs[4].

Notably, the photo-action spectra of MAPbBr$_3$ and MAPbI$_3$ nanocrystals in Fig. 1a, b show substantial carrier losses at higher excitation energies. In MAPbI$_3$ NCs, the PLQY drops from 25% when $\delta_E = 0.35$ eV to 18% when $\delta_E = 2.05$ eV, thereby leading to a loss of approximately 30% of the carrier conversion when high-energy excitations are employed. For MAPbBr$_3$ NCs, the PLQY plummets from 76 to 53% when $\delta_E$ increases from 0.45 eV to 1.25 eV. As reported in Supplementary Fig. 4, similar results were obtained by measuring the photoluminescence excitation spectrum and taking its ratio to the absorption spectrum. The pronounced losses even in the high-PLQY MAPbBr$_3$ NCs suggest that these effects are not correlated with the absolute value of the PLQY. Rather, these observations suggest that highly energetic excitations experience enhanced interactions with the trap sites, as compared with relaxed carriers. The implications of excitation energy-dependent PLQY in LHP nanostructures are two-fold: (i) imposing reconsiderations over the defect tolerance of LHPs; (ii) drawing attention to applications and devices that rely on the broadband absorber nature of PNCs. Indeed, potential applications of MAPbI$_3$ NCs in light-emission, lasing, and photovoltaics, as illustrated in Fig. 1a, rely on high-energy excitation and therefore are expected to experience lower conversion efficiencies.

Although the photophysics of MAPbI$_3$ and MAPbBr$_3$ NCs exhibit differences, the trends in the photo-action spectra are comparable: sharp decreases in PLQY (zone I) that ease off at higher $\delta_E$ (zone II). Figure 1c shows the absorption and photoluminescence (PL) spectrum of MAPbI$_3$ and MAPbBr$_3$ NCs, which fall in different ranges of the visible spectrum. Using the Elliott formula, we deconvolved the contributions from free carrier absorption, Rayleigh scattering, and excitonic absorption to the absorption spectra of PNCs, as shown in Fig. 1c[33,34]. A prominent excitonic feature in the MAPbBr$_3$ NCs spectrum can be observed in Fig. 1c, d, and from its higher exciton binding

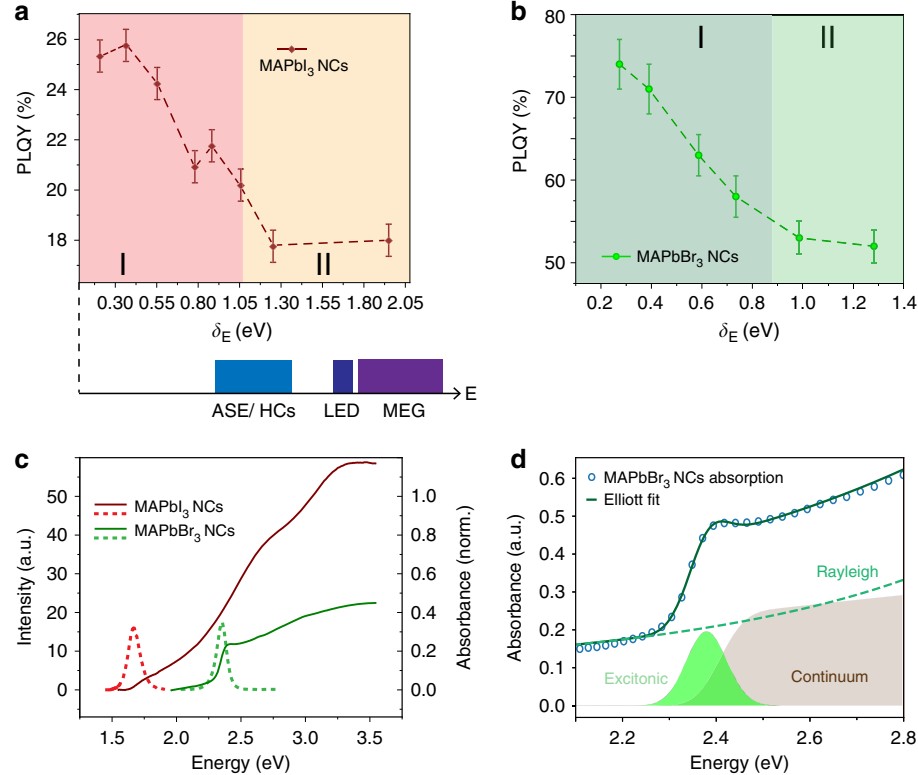

**Fig. 1 Steady-state signatures of hot carrier losses.** Excitation energy-dependent photoluminescence quantum yield spectra for (**a**) MAPbI$_3$ and (**b**) MAPbBr$_3$ NCs, reported as a function of excess excitation energy ($\delta_E$) with respect to the bandgap obtained from the fitting the absorption spectra with the Elliott formula (Table 1). Also known as 'photo-action spectra'. Red squares and green circles represent MAPbI$_3$ and MAPbBr$_3$ NCs PLQY values, respectively. Lines are guides for the eye. Error bars represent the standard deviation, estimated through repeated measurements. Excitation energy ranges for potential applications of MAPbI$_3$ NCs are included below Figure **a**. **c** Absorption (solid line) and photoluminescence (dashed line) of MAPbBr$_3$ and MAPbI$_3$ NCs in anhydrous toluene solutions. **d** Fitting of the MAPbBr$_3$ NCs absorption spectrum with the Elliott formula, deconvolving the contributions due to free carrier absorption continuum (beige region), Rayleigh scattering (green dashes), and excitonic absorption (bright green region).

**Table 1 Parameters for Elliott Model fitting of MAPbBr$_3$ and MAPbI$_3$ NCs absorption spectra.**

| Sample | Proportionality constant (a.u.) | Energy bandgap (eV) | Exciton binding energy, $E_b$ (meV) | Linewidth (meV) | Rayleigh scattering (a.u.) | Non-parabolicity (eV$^{-1}$) |
|---|---|---|---|---|---|---|
| MAPbI$_3$ | 0.37 ± 0.01 | 1.73 ± 0.01 | 20 ± 2 | 59 ± 3 | 13 ± 3 | 2.1 ± 0.1 |
| MAPbBr$_3$ | 0.46 ± 0.02 | 2.41 ± 0.01 | 36 ± 1 | 41 ± 1 | 6.8 ± 0.8 | 0.10 ± 0.01 |

energy (Table 1), higher than the thermal energy at room temperature ($k_B T$ of 26 meV at $T = 300$ K). Hence, we can infer the presence of stable excitons in MAPbBr$_3$ NCs.

On the other hand, the excitonic feature is much less pronounced in MAPbI$_3$ NCs (Fig. 1c, Supplementary Fig. 5). Weak excitonic contributions are present in the MAPbI$_3$ NCs linear absorption spectrum, likely to reflect strong electron-hole interactions rather than the presence of stable excitons. This is supported by the low-exciton binding energies, equal to or lower than the room temperature thermal energy ($k_B T$ of 26 meV), obtained from the Elliott fitting (Table 1) and consistent with the literature[35]. Despite the differences in the cold carrier dynamics, the dynamics of high-energy excitations are likely to be similar[36]. These excitations produce hot carriers rather than hot excitons due to momentum conservation constraints (i.e., low probability of indirect exciton absorption). Therefore, the early time dynamics is expected to be similar, and differences between MAPbI$_3$ and MAPbBr$_3$ NCs systems are likely to arise after the cooling and exciton formation[37,38].

**The role of hot carrier-induced trapping in perovskites.** To delve deeper into the origins of this loss of carriers, we investigated the fate of the hot carriers using femtosecond transient absorption (fs-TA), namely PP, and PPP spectroscopy. Building on previous works that studied the hot carrier dynamics[39] and transfer[40], we use the archetypal MAPbI$_3$ NCs as a model system for ultrafast spectroscopic measurements.

As shown in Fig. 2a, the fs-TA measurement of the MAPbI$_3$ NCs hot carrier cooling dynamics shows the characteristic broadening of the main photobleaching peak (PB, i.e., positive $\Delta T/T$), located at 1.70 eV[24,37,41,42]. Following past reports on hybrid and inorganic PNCs, we extracted the hot carrier temperatures and their cooling rate (Fig. 2a inset)[42]. In agreement with literature[42], two main cooling regimes are observed: in the low-fluence regime, the cooling is driven by carrier-phonon scattering, with a characteristic lifetime of $\tau_{cooling} = 500 \pm 100$ fs; in the high-fluence regime, an additional slower cooling mechanism arises, and it is related to the Auger re-heating processes taking place on the picosecond timescale. Hence, to

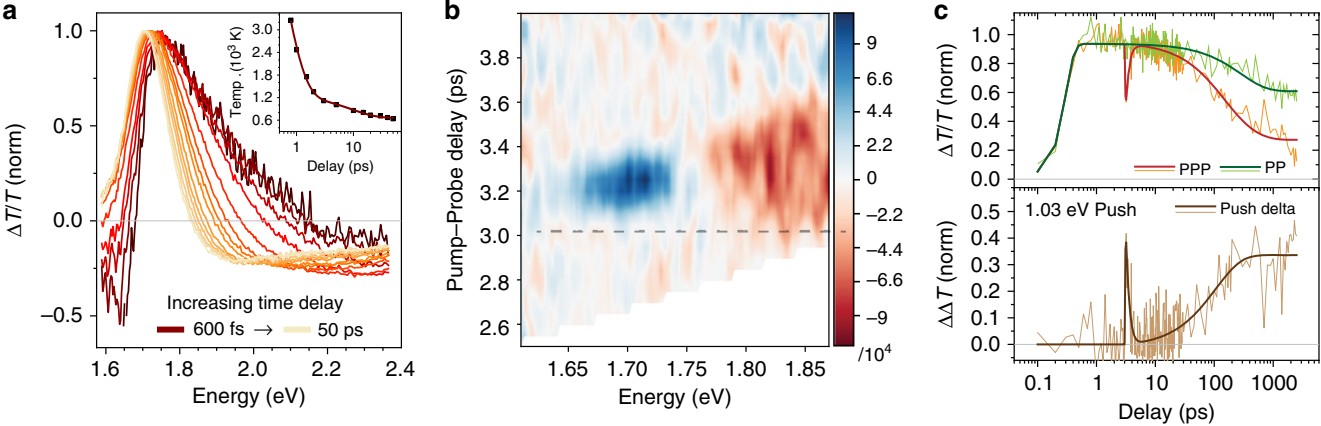

**Fig. 2 Time-Resolved Signatures of Hot Carrier-induced Trapping. a** Transient Absorption spectrum of MAPbI$_3$ NCs in anhydrous toluene solutions under intense (<$N$> = 3.6 e–h) high-energy excitation of 3.10 eV (around 1.4 eV in excess of the bandgap). Under this intense excitation, biexponential cooling dynamics is observed and reveals the coexistence of Auger and phonon-mediated relaxation pathways[24]. Inset: Corresponding hot carrier temperatures (black squares), estimated using the Boltzmann model. Multi-exponential fit is represented by the brown line. **b** Pump-push-probe differential spectrum of MAPbI$_3$ NCs in anhydrous toluene solutions, pumped at 2.07 eV (10 μJ cm$^{-2}$, <$N$> = 0.88), probed over the 1.60–1.90 eV interval, and pushed at 1.03 eV (1 mJ cm$^{-2}$). The black dashed line represents the push time-zero, the positive ΔΔT signal (blue) implies a decrease in the PB signal, and a negative ΔΔT signal (red) an increase in the PB signal. **c** Pump-probe (PP) and pump-push-probe (PPP) kinetics pumped at 1.91 eV (10 μJ cm$^{-2}$, <$N$> = 0.8) and probed at 1.70 eV, with push energy of 1.03 eV (1 mJ cm$^{-2}$). Thin lines represent the experimental data, and thick lines are exponential model fits to the data.

avoid complications that could obfuscate analysis and interpretation, we conducted our PPP experiments with low-pump fluence. Raw normalized PPP data are reported in Supplementary Figs. 19–23.

The chirp-corrected differential PPP spectrum for MAPbI$_3$ NCs with 1.03 eV push energy is shown in Fig. 2b, as the difference between PP and PPP spectra, i.e., where positive ΔΔT signal implies decreased PB, and vice versa. The reduced band-edge PB signal suggests that a fraction of the carriers at the band-edge absorbs the ultrafast push radiation (i.e., free-carrier absorption), before thermalizing and thereby re-heating the carrier distribution. Indeed, the presence of a negative signal at ~1.8 eV, blue-shifted with respect to the band-edge, is a signature of the heating of the carrier distribution analogous to the broadening of Fig. 2a. Thus, from PPP measurements, we can directly observe what happens to the hot carriers, and therefore we have the unique opportunity to investigate the effect of excess energy on the carrier dynamics with the presence of a push.

Figure 2c shows the comparison between the PP and the PPP kinetics of MAPbI$_3$ NCs with a push providing excess energy, $\delta_E$ = 1.03 eV. After the initial pump (2.07 eV), carriers relax to the band-edge and generate an intense PB signal at the band-edge position (probe 1.70 eV)[43]. A single-exponential behavior describes the kinetics of the band edge bleaching with lifetime $\tau_1$ = 4 ± 1 ns. Using a push energy of 1.03 eV (1 mJ cm$^{-2}$) at 3 ps PP delay time, the push pulse depletes a fraction of the band-edge population[39]. Subsequently, there is a complete recovery (or thermalization) of the excited carriers with a lifetime, $\tau_{thermal}$ = 450 ± 100 fs, in agreement with the hot carrier relaxation times observed from fs-TA spectroscopy. Although these push-generated hot carriers cool down completely (restoring the full PB signal), the subsequent dynamics experience a faster carrier recombination, compared to the case when the push is absent. This is also indicated by the growth of the PPP ΔΔT kinetic at longer time delays, with an additional lifetime component ($\tau$ = 110 ± 10 ps). As shown in Supplementary Figs. 6–8, this effect persists for different pump-push delay times and push fluences.

The hot carrier behavior for a high-energy push (2.07 eV, 60 μJ cm$^{-2}$) is entirely different and indicates a sudden loss of carriers when the push is absorbed (Supplementary Fig. 9). This

sudden disappearance of the carriers could indicate that at higher excess energies other carrier loss mechanisms are present (i.e., higher-lying traps), analogous to previously observed photoionization mechanisms[44]. Moreover, faster recombination dynamics after the push are also observed for the visible-push.

In principle, enhanced Auger recombinations from push-ground state absorption processes could cause analogous faster recombination dynamics after the push. For instance, an IR-push (1.03 eV) could generate an additional population from the push ground-state absorption (via two-photon absorption processes), which would be invisible to our detection scheme. In this case, the additional population could enhance the bimolecular and tri-molecular (Auger) recombination processes, thereby resulting in faster decay in our PPP kinetics. However, we quantified that this Auger contribution is minute for the case of our IR-push (Supplementary Note 1 and Supplementary Figs. 24–27). On the contrary, we demonstrate that the observed additional lifetime component arises from enhanced monomolecular recombinations (i.e., trapping).

On the other hand, the presence of push ground-state absorption is more significant when employing higher energy pushes (i.e., our visible 2.07 eV push), due to direct competition between the ground-state and the excited-state absorption processes. In this case, our analysis in Supplementary Note 1 shows that enhanced Auger dominates the recombination mechanism after the push. Hence, although the observed positive ΔΔT upon visible push absorption is a strong indication of possible trapping at high-lying trap sites, the presence of overlapping spectroscopic responses makes their complete disentanglement challenging. Therefore, we limit our further investigation to lower energy excesses.

The unusual behavior observed with increasing excess energy is also reflected in the excitation energy-resolved PP spectroscopy, as shown in Supplementary Fig. 10. When increasing $\delta_E$ = 0.4 to 0.8 eV by changing the pump energy, the bleaching lifetime reduces from 3.2 ± 0.6 ns to 1.9 ± 0.4 ns, respectively. Notably, when $\delta_E$ is further increased (i.e., at $\delta_E$ = 1.4 eV), the lifetime does not reduce further, indicating that a certain saturation exists. These results corroborate the PLQY photo-action spectra discussed earlier: when the carriers are residing in higher energy

states, relaxation and high-energy trapping are observed and provide a possible explanation for the lower PLQY at high $\delta_E$. Meanwhile, PPP spectra for MAPbBr$_3$ NCs show a fundamental difference from that obtained for MAPbI$_3$ NCs, i.e., no additional fast recombination components are detected—see Supplementary Fig. 11. We attribute this difference to arise from the nature of photoexcited species in these two systems. In the less excitonic MAPbI$_3$ NCs (dominated free carriers), the IR push can readily excite the free carriers, thus affecting the whole photobleaching band (Supplementary Figs. 12 and 14). This is in contrast with the more excitonic MAPbBr$_3$ NCs, where the differential PPP has a derivative-like shape, suggesting that inherent limitations of the PPP technique—possibly conceal the spectroscopic signatures of hot carrier-induced trapping (Supplementary Figs. 13, 14). Nonetheless, the hot carrier-induced trapping is still present in MAPbBr$_3$ NCs—evident from the PLQY photo-action spectra.

## Discussion

Herein, we propose a possible explanation for the observed dynamics in the PPP spectrum (IR push) that involves the interaction of hot carriers with shallow traps. Thus, while cold carriers mainly interact with deep traps as observed by non-unity PLQY values at very low-excess energies, we propose that the push-induced heating of the carriers opens additional trapping channels. Our hypothesis is based on the two main spectroscopic features arising from the PPP: (1) the ultrafast relaxation leading to a full recovery of the PB signal after the push pulse; and (2) the enhanced recombination induced by the push (Fig. 2c). These observations suggest that the hot carriers could promote the formation of shallow trapped carrier states, where one carrier (either the electron or the hole) gets trapped at a shallow trap site but still contributes to the signal in terms of defect stimulated emission. This would explain the observed faster recombination after the push as increased non-radiative recombination caused by shallow trap states[45,46]. Our results suggest that this trapping takes place on the same timescale as the cooling. Analogous considerations on HC trapping were reported in past studies on CdSe[47] and recent studies on MAPbI$_3$ thin films[19]. However, although the presence of shallow traps in PNCs is already well-known and provides a good explanation for their PL dynamics[48–51], the interactions of hot carriers with these traps have yet to be understood.

While PNCs surface chemistry shows some differences compared with that of conventional II-VI quantum dots (QDs) (e.g., CdSe), they both share similarities such as the presence of undercoordinated surface metal atoms, which act as carrier traps[29,52]. Hence, we use the well-established understanding of surface traps in II-VI QDs as our starting point. The pivotal role played by shallow traps in CdSe QDs was reported by Scholes and Kambhampati in their seminal papers, where the charge-trapping process to shallow defect sites is described using the Marcus (or Marcus-Jortner) charge transfer model[45,46]. The corollary of these works is that the trapping/detrapping equilibrium governs the population of charged dots (i.e., the number of trapped carriers in the ensemble), and this limits the PLQY due to faster non-radiative recombination for trapped carriers[17,32,49]. Notably, the same trapping dynamics has been reported for cold carriers in MAPbBr$_3$ and MAPbI$_3$ NCs, thus confirming the similar role played by surface traps[51,53–55]. Such an equilibrium—usually dictated by many factors, e.g., the temperature, the energetic levels of the traps—can be perturbed by the increased trapping rates for the hotter carriers.

Based on our qualitative understanding so far, we developed a model to describe the hot carriers and how they interact with these trap states in MAPbI$_3$ NCs (Fig. 3a). Briefly, for MAPbI$_3$

NCs the population dynamics of both the free carrier ($N$) and shallow-trapped carriers ($N_T$) states can be described by the following expressions:

$$\frac{dN(t)}{dt} = -k_2 N^2(t) - \int_0^\infty k_T(\epsilon)n(\epsilon,t)d\epsilon + e^{\Delta G_0/k_B T}\int_0^\infty k_T(\epsilon)n_T(\epsilon,t)d\epsilon, \tag{1}$$

$$\frac{dN_T(t)}{dt} = -k_{tr}N_T(t) + \int_0^\infty k_T(\epsilon)n(\epsilon,t)d\epsilon - e^{\Delta G_0/k_B T}\int_0^\infty k_T(\epsilon)n_T(\epsilon,t)d\epsilon. \tag{2}$$

Here, $\epsilon$ is the carrier energy axis; $k_2$ is the bimolecular recombination rate; and $k_{tr}$ is the non-radiative recombination rate from the trap potential; $n(\epsilon, t)$ is the thermally equilibrated carrier distribution with temperature $T(t)$ in a parabolic band system; $k_T(\epsilon)$ is the population exchange rate between the two potential energy surfaces as a function of carrier energy; and $\Delta G_0$ is the Gibbs free energy (i.e., the difference in energy between reactant and product energy surfaces at the equilibrium). The second and third terms in Eqs. (1) and (2) describe the population exchanges between the parabolas. We assume the phenomenological carrier temperature to be $T(t) = 300 \, K + T_0 \exp(-t/\tau_c)$, where $T_0 = \delta_E/k_B$ is the carrier temperature post-excitation (assuming the almost instantaneous formation of Boltzmann distribution), that cools down with cooling time constant $\tau_c$[24]. In agreement with Elliott fitting results, we accounted for the more excitonic nature of the recombination in MAPbBr$_3$ NCs by replacing the carrier population term presented in Eqs. (1) and (3). Specifically, the radiative bimolecular recombination term is replaced by a monomolecular radiative recombination term with rate $k_r$, [i.e., $-k_2 N^2(t) \to -k_r N(t)$]. The full description of the model is given in Supplementary Note 2. The PLQY of the system with an initial photoexcited carrier population $N_0$, can be calculated as:

$$\text{PLQY}(\delta_E) = \frac{k_2 \int_0^\infty N^2(t)dt}{N_0}. \tag{3}$$

It is worth noting that in this model the hot carriers have two competing pathways (Fig. 3b, c): (1) relaxation to the band-edge, in accordance with the literature on MAPbI$_3$[56]; and (2) being trapped at the traps' potentials. The model also describes how the trapping of hotter carriers occurs with a higher driving force, $-(\Delta G_0 + \delta_E)$, compared with the cold carriers. Therefore, trapping rates increase and become more dominant as carriers get hotter (due to the $\delta_E$ term), thereby leading to a higher number of trapped carriers within the PNCs.

This can explain the downward PLQY dependence with increasing excess energy, and the effects are clearly illustrated in Fig. 3d, e. These findings have very strong implications on the impact of the traps on hot carriers: the defect tolerance[29] declines when carriers are more energetic. In other words, these traps become activated under high-energy excitation conditions, a hallmark of applications such as light-emitting devices and photovoltaics.

To gain a deeper insight into the surface chemistry of the defects in PNCs, we exchanged the native alkylamine/oleic acid ligand couple with trioctylphosphine oxide (TOPO). TOPO has previously been shown to passivate and improve the emissive properties of bulk MAPbI$_3$ thin films[57,58]. MAPbI$_3$ and MAPbBr$_3$ share common defect chemistry, and a remarkably defect tolerant electronic structure (i.e., the ability to retain the electronic structure even with large defect concentrations)[7,27,29,59]. For instance in MAPbI$_3$, many first-principle calculations, and experiments (PL, XPS) show that most of the stable defects (i.e., vacancies $V_{MA}$, $V_I$, $V_{Br}$, and interstitials $MA_i$, $Pb_i$) introduce shallow traps, while deep electron/hole traps are associated with

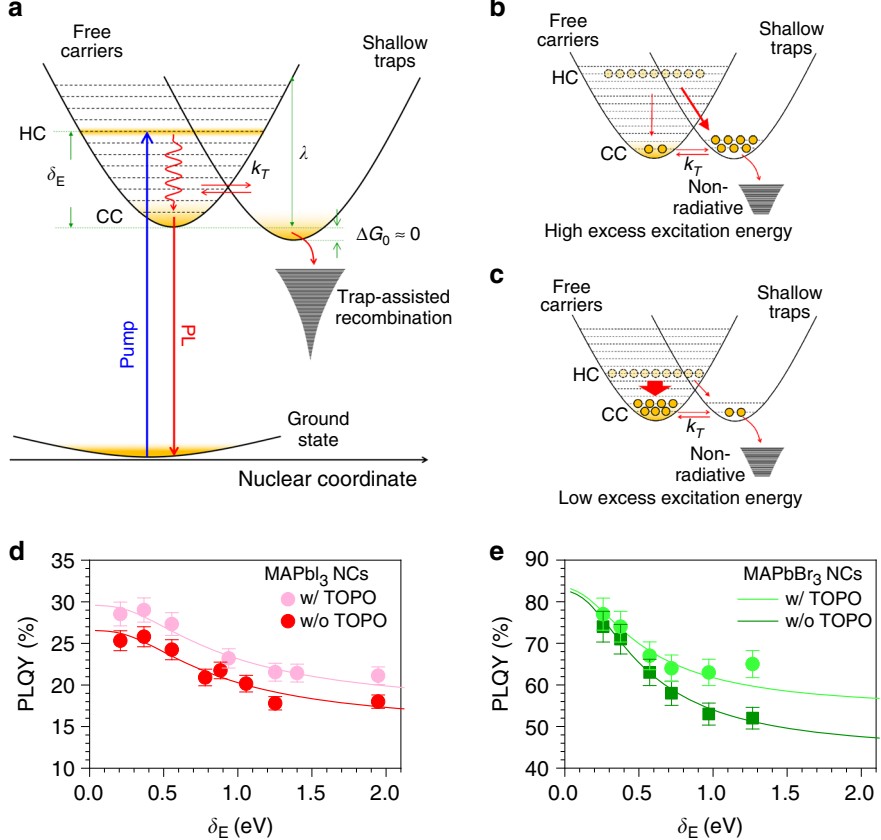

**Fig. 3 Phenomenological model of hot carrier-induced trapping. a** Model schematic describing the interaction between carriers and shallow traps in PNCs. Free and trapped carriers are involved in a thermal equilibrium, whose barrier is dictated by the combination of reorganization energy, $\lambda$, and trapping free energy $-\Delta G_0$. **b** The high-excess energy ($\delta_E$) for the carriers increases the trapping rate and hence results in a higher number of trapped carriers, i.e., lower PLQY. **c** Vice versa, lower $\delta_E$ results in lower trapping rate and thus results in a lower number of trapped carriers, i.e., higher PLQY. HC and CC indicate the hot carriers and cold carriers, respectively. Model fitting of the photo-action spectra, for (**d**) MAPbI$_3$ NCs (**e**) MAPbBr$_3$ NCs. The light-colored dots represent TOPO ligand-exchanged PNCs, while dark dots represent pristine PNCs. Error bars represent the standard deviation, estimated through repeated measurements. The corresponding colored lines represent the model fitting.

the less abundant interstitial iodine (I$_i$) defects and lead vacancies (V$_{Pb}$)[6,7,59]. Furthermore, the electronic structure was found to be particularly stable with respect to the formation of charge compensating defect pairs (e.g., V$_{MA}$ and V$_I$)[29]. According to the Green taxonomy, the P = O moiety makes TOPO a neutral donor (L-type ligand) that can efficiently passivate under-coordinate metal (Pb) atoms from these vacancies through Lewis adduct formation[52,60,61].

Our model quantifies the hot carrier-trap interaction in terms of the two most important parameters: $H_s$, the electronic coupling of the carriers to the defects; and $\lambda$, the reorganization energy associated with the trap-induced structural distortions[46]. The values of $H_s$ and $\lambda$ (Supplementary Table 1) are consistent with previous reports of CdSe QDs[46], and they are reduced when TOPO-passivation is applied. As shown in Fig. 3c, TOPO-passivation is beneficial for PNCs, providing overarching improvements in the PLQY of MAPbI$_3$ NCs, and mitigating the loss in PLQY at high-excess energies for MAPbBr$_3$ NCs ($\delta_E$ > 0.2 eV). As the excitation energy is closer to the band-edge, shallow traps are essentially optically-inactive, and the observed improvement is smaller. With increasing excess energy, improper passivation of the shallow traps, or the lack thereof, consistently leads to a loss of carriers relaxing from high-energy states. Our PPP results corroborate these findings and are reported in Supplementary Figs. 16, 17. Here, the comparison between differential PPP kinetics for pristine and ligand-exchanged MAPbI$_3$

evidences how the additional TOPO passivation results in lower carrier losses to both ligands and shallow traps sites. Therefore, these experiments confirm that the observed reduction of the PLQY is related to the surface passivation of the perovskite NCs. Specifically, traps that are supposedly benign or 'inactive' for the cold carriers, and whose passivation is generally not considered as necessary[7], become active under the energy (or the 'heat') of the hot carriers.

Hot carrier-induced trapping processes provide an additional and alternative cooling pathway that intuitively should speed up overall hot carrier cooling times. However, a direct comparison between PPP data for pristine and ligand-exchange MAPbI$_3$ NCs (Supplementary Fig. 16b) indicates that while the long-time recombination dynamics are affected by the passivation, the cooling dynamics are unchanged within our time resolution. Similar results for the cooling were recently reported by Harel et al. for bulk MAPI and Bakulin et al. for other PNCs[19,62]. Hence, we speculate that defects passivated by TOPO do not significantly affect the electron-phonon coupling, and therefore do not significantly speed up the cooling dynamics.

Nevertheless, a proper passivation is crucial for HC solar cells, ASE and lasing applications. Indeed, as suggested by the passivation effect of HC temperatures (Supplementary Fig. 16e), HC carrier losses could still affect these applications in terms of their efficiencies (i.e., reducing the number of carriers available for extraction or photon conversion, respectively)[4]. Future efforts in

studying the passivation effects on these applications are needed. Moreover, the increased knowledge of HC-induced trapping processes provides a different perspective on the many unanswered questions in the perovskite field, e.g., polaron dynamics, hot carrier cooling and extraction, and multiple exciton generation (MEG). Zhu and co-workers have provided an excellent picture of the hot polaron formation dynamics, as an alternative relaxation pathway to carrier-LO phonon scattering[62]. Notably, the study of hot polarons is usually conducted at low fluences below the Mott Density, which is also the regime of interest in this study[63]. It is therefore interesting to note that these hot carrier-induced trapping processes might complete the picture of large polarons[64] in PNCs, and could help rationalize the absence of hot PL in CsPbBr$_3$[65,66].

In conclusion, we uncover the detrimental effects of hot carrier traps on the PLQY of MAPbBr$_3$ and MAPbI$_3$ PNCs. Our excitation energy-dependent PLQY and PPP measurements demonstrate that when hot carriers with moderate energies in excess of the band-gap are present, shallow trap sites are activated. This startling activity of shallow defects, previously deemed as innocuous, suggests a reconsideration of the defect-tolerance in these systems. We use a phenomenological model to explain why hotter carriers experience increased trapping rates to low-energy shallow traps, thereby transforming these benign defects into malignant ones. Our model also reveals how the electronic coupling of the carriers to the traps and its associated reorganization energy can be changed by passivating the PNCs' surface. We experimentally support this by performing a ligand exchange (with TOPO as the ligand) to passivate these traps, leading to tangible improvements in the PLQY excitation profile. However, the adopted procedure is but one of many possible strategies that can be employed, and we are confident that these insights will help pave the way towards highly efficient PNC devices. Our findings provide a more holistic understanding of 'defect-tolerant' PNCs and have potential implications on how the future photophysical studies of high-energy carriers can be interpreted.

## Methods

**Materials**. Lead Bromide, PbBr$_2$ and Lead Iodide (99.999%, trace metals basis), oleylamine (>98%; OlAm), n-octylamine (99%; OctAm), oleic acid (90%: OlAc), N,N-dimethylformamide, (99.8% *anhydrous*; DMF), acetonitrile (99.8% *anhydrous*; ACN), toluene (anhydrous, 99.7% GC), trioctylphenylphosphine (99% *Reagent Plus*; TOPO®) and benzyl alcohol (99.5%, BnOH) were purchased from Sigma-Aldrich. Methylammonium bromide (MABr) and methylammonium iodide (MAI) were purchased from Greatcell Solar Material. The chemicals were used without further purification.

**Synthesis of MAPbBr$_3$ NCs**. The precursor solution was prepared in 2 mL DMF, dissolving completely 0.04 mmol of PbBr$_2$ (146.8 mg) and 0.04 mmol of MABr (44.8 mg). After the complete dissolution of the salts by bath sonication, the PNCs were synthesized by swift injection of 200 μL of precursor solution into an anti-solvent solution under vigorous stirring. The antisolvent solution was prepared adding 20 μL OctAm, 1000 μL OlAc, and 700 μL to 5 mL of toluene. After the swift injection, a yellow/green suspension was obtained. The purification of the crude followed previously a previously published synthesis (ref. [18], main text) and comprised two centrifugation-redispersion steps at 12,000 rpm and 4000 rpm, respectively. Subsequently, the precipitated nanocrystals were dispersed in 1 mL of anhydrous toluene and stored in at 4 °C.

**Synthesis of MAPbI$_3$ NCs**. The precursor solution was prepared by dissolving completely 0.1 mmol of PbI$_2$ (45.9 mg) and 0.1 mmol of MAI (15.9 mg) into a solution of 2 mL ACN, 20 μL OlAm, 0.20 mL OlAc. The full precursor solution was slowly injected (dropwise addition) to 10 mL toluene vigorously stirred under an ambient atmosphere. The purification of the crude comprised one centrifugation/redispersion step at 8000 rpm and a second centrifuge step at 4000 rpm to remove the bulkier nanoparticles. The supernatant was then stored in a nitrogen glovebox for further usage.

**X-ray diffraction (XRD)**. XRD measurements on drop cast thin film samples were performed using Rigaku SmartLab High-Resolution X-ray diffractometer with Cu Kα (wavelength of 1.5418 Å) X-ray source. The measurement was done in the general medium resolution PB/PSA (theta-2 theta) mode scanning from 10 to 45 degrees with steps of 0.02 degrees.

**Linear optical spectroscopy**. UV-Vis spectra were recorded by using a Shimadzu UV-3600 Plus spectrometer, in the range 200–900 nm. The PL spectra were measured with a FluoroLog (Jobin-Yvon) fluorimeter. Diluted sample solutions in anhydrous toluene were contained in quartz cuvettes.

**Photoluminescence quantum yield (PLQY)**. PLQY measurements were performed using a Horiba Jobin-Yvon Fluorolog system equipped with iHR320 monochromator, coupled with a photomultiplier tube and a spectrally calibrated Spectralon-coated integrating sphere (Quanta-Phi). Excitation energy was varied by selecting different components of a Xe lamp emission with a monochromator. Diluted solutions of the samples (OD < 0.1) were contained in a quartz 1 cm × 1 cm cuvette. A relative error of about 5% on measured PLQY values was estimated by repeated measurements.

**Femtosecond transient absorption (fs-TA) spectroscopy**. fs-TA measurements were performed using a Helios spectrometer (Ultrafast Systems, LLC). The pump pulse was a frequency-doubled fundamental emission (400 nm, 3.1 eV pulse) generated from a Coherent Legend (150 fs, 1 kHz, 800 nm) regenerative amplifier. Excitation energy-resolved pump pulses were generated from an optical parametric amplifier (Light Conversion TOPAS-C). The white light continuum probe pulse (in the range from 420 to 780 nm) was generated by focusing the residual of the regenerative amplifier's fundamental 800 nm laser pulses into either a 2 mm sapphire crystal (for visible range). Multichannel acquisition of the entire spectrum was achieved using a cMOS detector. Samples were contained into 2 mm cuvette and vigorously stirred using a Magnetic Stirrer (Ultrafast Systems, LLC) to avoid photocharging effects. The calculation of the absorption cross section for MAPbI$_3$ NCs is reported in Supplementary Fig. 15.

**Pump-Push-Probe (PPP) spectroscopy**. A home-built setup in transmission geometry was used to perform PPP spectroscopy measurements. The output from a Coherent Libra regenerative amplifier (1 kHz, 800 nm, 50 fs) was split to three beams, two to pump two Coherent OPerA-Solo optical parametric amplifiers. The remaining 800 nm fundamental beam used for WLC generation (450 to 780 nm) was attenuated with a 750 nm short-pass filter (estimated probe fluence of ~0.14 μJ cm$^{-2}$ at ~1.63 eV). The first OPA was used to generate the pump pulse train (1.91 eV, 2.07 eV), while the second OPA was used to generate the push pulse train (1.03 eV, 2.07 eV). In the experiments on MAPbBr$_3$ NCs, the 3.1 eV pump was generated by the frequency-doubling of the residual fundamental output using a BBO crystal. The pump was chopped at 83 Hz, in combination with a modulated push, the PP, and PPP signals were obtained separately and averaged across at least 3 scans. Taking the difference yields the push-induced signal (described in the main text). Both push and probe pulse trains were mechanically delayed by precision delay stages and the probe was collected by a spectrometer (Princeton Instruments Acton SP-2300i) coupled to a PMT point detector and collected by the computer through an SRS 830 lock-in amplifier. Normalized data is presented in the manuscript, while raw data is reported in the SI. The stability of the samples during PPP experiments is given in Supplementary Fig. 18.

## Data availability
The data that support the findings of this study are openly available in DR-NTU (Data) at: https://doi.org/10.21979/N9/EGH6UI. Data is also available from the Corresponding Author upon reasonable request.

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

## Acknowledgements

This research was supported by Nanyang Technological University under its start-up grant (M4080514) and its JSPS-NTU Joint Research Project (M4082176); by the Ministry of Education under its AcRF Tier 2 grants (MOE2016-T2-1-034 and MOE2017-T2-2-002); and by the National Research Foundation (NRF) Singapore under its NRF Investigatorship (NRF-NRFI-2018-04).

## Author contributions

Conceptualization, M.R., S.S.L., D.G., T.C.S.; Methodology, D.G., J.W.M.L.; Investigation, M.R., S.S.L., D.G.; Resources, M.R., Q.Z., S.R., E.Y.K.T.; Writing – Original Draft, M.R., Writing – Review & Editing, M.R., S.S.L., D.G., T.C.S.; Visualization, M.R., S.S.L., D.G., J.W.M.L., Supervision, T.C.S.; Funding Acquisition, T.C.S.

## Competing interests

The authors declare no competing interests.
