## [Peer Review File · Nature Communications]

Reviewers' comments:

Reviewer #1 (Remarks to the Author):

The manuscript by M. Righetto et al. discusses the details of hot carrier relaxation and the influence of the trapping centers on the photoluminescence quantum yield (QY) of MAPbX₃ perovskite nanocrystals. Authors employ steady state QY measurements and femtosecond pump-probe (PP) and pump-push-probe (PPP) spectroscopy as well as kinetic equation modeling to quantify the influence of excess excitation energy (δE) on QY.

The concept of the measurements and the results of this work are clearly presented. The manuscript reads well and discussion is presented without any hindrances/avoidances. Although the effect of excitation wavelength on the QY of nanocrystals, including perovskites, has been sporadically and intermittently discussed, this work presents the first comprehensive study that shows clear influence of δE on the emission properties of perovskite NCs. Thus, the manuscript presents an important advance in the field and will be relevant to the majority of researchers working in the field of perovskites. At the same time, I feel that several important questions need to be resolved before the manuscript could be considered further.

1) Authors employ PP spectroscopy at different excess energies, which is not the easiest thing to do. Why not record time-resolved PL under the same conditions? Whenever the hot carriers experience the increased scatter/loss, it should result in faster recombination kinetics. I'm also concerned that using high fluences in push experiment (up to 1.7 mJ/cm²) may result in some unwanted artifacts and/or sample degradation. Using time resolved PL that allows many orders of magnitude lower pump fluences will be much safer in terms of sample degradation for which perovskites are well known.

2) Discussion about PPP displayed in Figure 2(c,d) should be better articulated. It is not clear why the push at lower $\delta E = 1.03$ eV in panel (c) results in the appearance of a faster component in kinetics while at the same time, push with larger energy $\delta E = 2.07$ eV in panel (d) seems to provide same decay components. I would also suggest having dynamics without normalization because it is unclear how much signal is lost from the regenerated hot carriers when they get trapped.

3) For MAPbI₃ NCs, authors reach $\delta E > 2$ eV, which is more than twice the bandgap energy. Recent work has shown carrier multiplication (CM) in CsPbI₃ NCs, (de Weerd et al., Nat. Comm. (2018) 9:4199 | DOI: 10.1038/s41467-018-06721-0) at such excess energy. Authors should corroborate on how their results might relate and could CM contribute to the drop in PLQY at higher energies.

4) Authors develop modeling based on Marcus theory with corresponding rate equations. As it looks, it is entirely phenomenological model. It fits the observed kinetics based on rate constants, reorganization energy and coupling energy. However, as any other such model, it cannot predict the nature of the trapping centers. I think it is very important to develop first-principle model that can do that. As an example, it has been established via theoretical modeling that Br-vacancy centers are responsible for changes in PL in CsPbBr nanoparticles. As a result, high PLQY samples have been developed. Given a sufficiently large range of parameters in the phenomenological models, it is often possible to provide fitting to nearly every observable kinetic by simply adding more free parameters, which may or may not be of physical sense.

5) There is a confusion about the first term in Eq 1. It shows N^2 which is true for radiative recombination involving free carriers. In the text authors mostly refer to excitonic behavior and it makes things confusing. Only reading Suppl. Info it becomes more clear that for different samples (iodide vs. bromide) different terms are used. Please clarify it in the main text.

I think that absence of the first-principle calculations of defect formation is the main weakness of this work and should be addressed before the publication is possible. The authors themselves do experiments with TOPO ligands, which is a common substance for NC passivation and observe modifications. However, without the predictive model, further ligand modification seems to be difficult and would simply result in a trial and error approach.

Reviewer #2 (Remarks to the Author):

Sum and co-workers have studied the defect trapping process in perovskite semiconductor nanocrystals using the pump-push-probe measurements, and claimed the observation of excitation photon energy-dependent carrier trapping dynamics. They reported that the trapping rates of hot carriers increase with increasing the excess energy of excitation photons, which is proposed to be the origin of excitation energy dependence of PL QYs. This study could make an important contribution to the field studying perovskite materials meriting publication in Nature Communications if the conclusion stands. Nevertheless, several key issues need to be addressed.

1) My major concern is about the validity of the photon energies of push beams in the pump-push-probe measurements. Normally, the push photon energy is selected in the absorption band of the concerned excited states and to avoid the optical absorption of the ground states (see Richard Friend's works). In this study, the push beam was assumed to excite the carriers to higher excited states. However, the photon energies and excitation fluences of push beams can effectively excite the carriers from ground states:

a) Excited state absorption can be found at push photon energy of 2.07 eV (Fig. 2a); however, the ground state absorption is very strong (Fig. 1c).

b) The transient absorption spectrum in the range close to 1.03 eV was not reported in this work, so it remains unclear if the excited state carriers absorb the push beam. However, the multi-photon absorptions of the nanocrystals are highly efficient (e.g, a report from the same group, Chen et al., Nat. Commun. 8,15198(2017)). Ground state excitation cannot be avoided by the push beam since the push fluence is very high (1 mJ/cm²).

Since the direct excitation from ground states cannot be excluded, the current analysis only considering the excited state absorption is yet to be justified. In current configuration, it is possible to induced more than one electron hole pairs in a singlet dot by combining the excitations from pump and push beams. If so, the dynamics change in the 100s ps can be simply ascribed to Auger process rather than the assignment in this manuscript. It is essential to address this issue since the result of pump-push-probe measurement is pivotal for the conclusion in this work.

2) The diagrams of potential energy surfaces in the model (Fig. 3) are problematic. It is hard to image that the PESs of hot and cold carriers are not crossed. In current pictures in Fig.3, most of hot carriers will transfer to the trapped carries, which certainly is not consistent with the experimental result. I'd suggest that the authors carefully review the available knowledge about the band structure of the materials and make the model diagram more realistic. In my opinion, temperature-dependent measurements are highly preferred to justify the proposed model if applicable (optional).

3) In addition to the PLQYs at different excitation photon energies, the spectra of PL excitation should be included which, in principle, can better reflect the excitation energy dependences of PL emission in the samples. Please also check the stoichiometric ratios of the samples studied. If miscellaneous phases and structures with higher bandgaps (PbI₂, or other low dimensional perovskites) exist in the samples, the lower PL QYs with higher excitations can be naturally explained.

4) Optical stark effect is known to be important in these samples. Could the short-lived changes in Fig. 2b be relevant?

Moreover, some minor points should be carefully addressed.

5) Table 1, "Linewidth (eV)" seems to be "meV"? Figure S8 label for x-axis should be "ns"?

6) Most of the traces are plotted in normalized scale. I'd suggest the authors to include the exact amplitudes of the signals for references by future studies in the community.

Overall, this work is interesting and valuable for the community. I'd recommend publication of this manuscript if my major concern can be properly addressed.

Reviewer #3 (Remarks to the Author):

The manuscript by Righetto and coworkers reports a spectroscopic study of hot carrier trapping in MAPbI₃ and MAPbBr₃ perovskite nanocrystals using a combination of "pump-push-probe" spectroscopy and PLQY measurements. The results are rationalized in the framework of the Marcus charge transfer model.

I believe the overall topic and approaches in the paper are interesting to the perovskite community, but I have several concerns which I feel should be addressed before it can be published in Nature Communications.

1.

There is very little discussion about previous works on the role of defects and overall material quality on hot carriers in perovskites. How do the authors reconcile their observations with previous reports, especially those that indicate the absence of hot carrier trapping (i.e. J. Phys. Chem. C 2017, 121, 21, 11201-11206, DOI: 10.1021/acs.jpcc.7b03992)?

2.

Can the authors comment more on (i) the timescale for (hot) carrier trapping, (ii) the energy of the trap states in the studied systems and (iii) whether hot and cold carriers are affected by the same traps?

3.

Compared to 1-sun illumination, transient optical experiments are performed at relatively high intensities. Can the authors comment on the significance of hot carrier trapping under the standard operational conditions of a device?

4.

The manuscript focuses more heavily on the iodide systems. The general consensus in the literature is that the iodide-based perovskites are more unstable (less defect tolerant) than the bromide equivalents. Based on the data in the manuscript, is hot-carrier trapping more prevalent in MAPbI₃ NCs than MAPbBr₃ NCs? For instance, the bromide samples exhibit a higher overall PLQY (Figure 1a+b) and a lack of significant push-induced changes in Figure S9.

5.

My most serious concerns are about the execution and interpretation of the pump-push-probe experiments.

a) In the methods, the authors should explain how the 3.1 eV pump is generated for the experiments on the MAPbBr₃ NCs.

b) Can the authors provide more explanation for the differences in Figure S5? Namely, (i) could the different timescales be related to residual hot carriers and phonons from the initial pump? This might have implications for other data in the manuscript. (ii) What happens to the late-time dynamics when

the push delay is increased?

c) The PP vs PPP differential transmission data are all normalized, and there is no information regarding the modulation of the push beam. This calls into question the interpretation of the push-induced signals.

d) The push pulses are quite intense ($\sim \text{mJ cm}^{-2}$). Do the authors observe any sample degradation (difficult to see with normalized data) under these conditions, and could this influence the overall carrier dynamics?

e) The "Push Delta" signals in Figure 2c+d look very different (the latter exhibits no ultrafast decay (carrier thermalization) after the push event). When pushing these MAPbI₃ NCs (1.73 eV bandgap) at such a high energy (2.07 eV), how can the authors possibly distinguish between the contributions from (i) the intended "heating"; of the excited (band-edge or trapped) states (ii) excitation into some higher energy (possibly "dark") electronic band, and most importantly (iii) excitation from the ground state?

f) As noted in (d), the push intensities are quite high. Following from point (e), in the case of the sub-gap push energies, is it possible that multi-photon absorption (as in their own work from ref. 22) can contribute to the optical response in PPP?

To help address points c-f, I recommend that the authors perform "push-probe"; control experiments (like pump-probe, but using a chopped push as the excitation beam) and compare this data with the PPP responses.

To conclude, I think the manuscript would substantially benefit from additional discussions and relatively simple control experiments outlined above. On a lesser note, there are some grammatical errors throughout the manuscript, but particularly in the abstract and introduction. For instance "its"; as opposed to "their" defect tolerance in the first line of the abstract, and "consist of predominantly of shallow traps"; on page 3. Overall, I do not believe that the manuscript, in its current form, meets the standard of Nature Communications, and would recommend seeking publication in a more specialized journal such as JPCL,

Reviewer #2

Sum and co-workers have studied the defect trapping process in perovskite semiconductor nanocrystals using the pump-push-probe measurements, and claimed the observation of excitation photon energy-dependent carrier trapping dynamics. They reported that the trapping rates of hot carriers increase with increasing the excess energy of excitation photons, which is proposed to be the origin of excitation energy dependence of PL QYs. This study could make an important contribution to the field studying perovskite materials meriting publication in Nature Communications if the conclusion stands. Nevertheless, several key issues need to be addressed.

We are grateful to Reviewer #2 for providing positive and constructive scientific feedback.

1) My major concern is about the validity of the photon energies of push beams in the pump-push-probe measurements. Normally, the push photon energy is selected in the absorption band of the concerned excited states and to avoid the optical absorption of the ground states (see Richard Friend's works). In this study, the push beam was assumed to excite the carriers to higher excited states. However, the photon energies and excitation fluences of push beams can effectively excite the carriers from ground states:

a) Excited state absorption can be found at push photon energy of 2.07 eV (Fig. 2a); however, the ground state absorption is very strong (Fig. 1c).

We thank the Reviewer for raising a valid point. This issue is also raised by Reviewer #1 (comment #2). For the sake of the coherence of this response letter, we will address this issue in the later part of this response on page 18.

b) The transient absorption spectrum in the range close to 1.03 eV was not reported in this work, so it remains unclear if the excited state carriers absorb the push beam.

We thank the Reviewer for raising this point. Following his/her suggestion, we have performed a broadband transient absorption study covering the IR region of interest (**Figure R2.1**). The result shows clearly the presence of a broadband excited state absorption across the IR region, which could be assigned to free carrier absorption from the band edge. This clarifies the concern raised by the Reviewer: the push beam can indeed be absorbed by carriers in the excited state.

Figure R2.1 The (a) surface plot and (b) time slice at 5 ps of the broadband transient absorption spectrum of MAPbI₃ colloidal nanocrystals (NCs) sample, pumped at 400 nm (3.10 eV at 42 μJ/cm²). The data shows a negative ΔT/T signal at 1.07 eV, which implies the presence of excited state absorption.

- We added this result in our revised the Supplementary Information as part of the *Supplementary Note 1 – Effects of Ground State Push Absorption*.

However, the multi-photon absorptions of the nanocrystals are highly efficient (e.g, a report from the same group, Chen et al., Nat. Commun. 8,15198(2017)). Ground state excitation cannot be avoided by the push beam since the push fluence is very high (1 mJ/cm²). Since the direct excitation from ground states cannot be excluded, the current analysis only considering the excited state absorption is yet to be justified. In current configuration, it is

possible to induced more than one electron hole pairs in a singlet dot by combining the excitations from pump and push beams.

We are grateful to the Reviewer for this thoughtful comment. Indeed, there could be a certain amount of carriers excited by ground state absorption of the push. Our reply to this comment is structured into two parts: i) we demonstrate that our instrumental setup is not directly sensitive to push ground state absorption; ii) we quantify the two-photon absorption (2PA) process to demonstrate that it does not affect our conclusions.

i) We refer to our recent publication (Lim *et al.*, *Sci. Adv.* 2019; 5:eaax3620), where we presented in detail our experimental setup. The schematic in **Figure R2.2** is replicated here for convenience.

Figure R2.2: Schematic of the pulses sequence in the pump-push-probe experiment. In our setup, the pump is modulated by a chopper, while a mechanical shutter modulates the push.

As illustrated here, our measurements are based on lock-in detection, operating at half of the pump frequency, which allows measuring the pump-probe signal. The pump-push-probe measurements are obtained by introducing a third pulse by means of a mechanical shutter. In this setup, the PPP signal is obtained as the difference between the pump-probe signal with and without the push. Therefore, with reference to **Figure R2.2**, the signal that we obtain is described as:

$$\begin{aligned}\Delta T_{\text{push off}} &= C - D \\ \Delta T_{\text{push on}} &= A - B \\ \Delta\Delta T &= \Delta T_{\text{push off}} - \Delta T_{\text{push on}}\end{aligned}$$

Within this detection scheme, it is clear that the effect of the direct excitation from the push is canceled out when performing the $A - B$ operation. However, the presence of an additional population excited by the push and not directly detected by the lock-in detection could still influence the observed dynamics. In other words, the Reviewer is correct about the possible presence of carriers photoexcited directly from the ground state by the push pulse, but they cannot be detected directly within our detection scheme. Therefore, the positive $\Delta\Delta T$ signal in our measurement comes directly from the free-carrier absorption. However, while these carriers are invisible to our direct detection, they could indeed still influence the observed overall dynamics. Notably, as will be explained in the latter part, it does not affect our interpretation.

Figure R2.3 Transient absorption kinetics at the band-edge (1.72 eV) of our MAPbI₃ colloidal NCs, pumped with 2.07 eV (at 10 μJ/cm²) and 1.03 eV (at 1 mJ/cm²). Comparing the population, the IR pump generates about 20% of carriers of those generated by the visible pump.

ii) To quantify the effect from this multi-photon absorption from the ground state, we performed TA measurement using two different pump energies: 2.07 eV (at $10 \mu\text{J}/\text{cm}^2$) and 1.03 eV (at $1 \text{ mJ}/\text{cm}^2$), which are analogous to the fluences reported in the Main Text for pump and push, respectively. These two pumps provide a comparison of the number of carriers excited from the ground state by our pump and push pulses during the reported pump-push-probe experiment. The result is shown in **Figure R2.3**.

If we consider the peak intensity of the signals, we can observe that at the PPP experiment particular fluence, the 1.03 eV pump will generate $\sim 20\%$ of carriers, compared to those which are generated by the 2.07 eV pump. This is not observed in terms of $\Delta\Delta T$ values during the push since our PPP technique is insensitive to this additional population. Hence, our observation of positive $\Delta\Delta T$ values (i.e., drop in ΔT values after push) is due to purely excited state absorption processes. These additional $\sim 20\%$ ground state excitation from the push, however, will not affect our current interpretation of the matter, as will be discussed in the next comment.

If so, the dynamics change in the 100s ps can be simply ascribed to Auger process rather than the assignment in this manuscript. It is essential to address this issue since the result of pump-push-probe measurement is pivotal for the conclusion in this work.

The Reviewer raises a very valid concern, however, as mentioned above, the IR push at the employed fluence can generate only up to $\sim 20\%$ additional carriers from the ground-state multi-photon absorption, with respect to the initial carrier population generated by the pump. To address the question whether the change in the dynamics after push relaxation can be ascribed to Auger processes in a quantitative way, our strategy is to first quantify the monomolecular, bimolecular, and Auger rates in our system; and then to estimate how much

these additional carriers from ground-state absorption would change the dynamics. These rates, together with a direct estimate of the absorption cross-section (Figure S15 of the revised SI), were determined via fluence dependent pump-probe measurements. Here, the population dynamics in the NCs can be simply modeled by the following rate equation:

$$\frac{d}{dt}\langle N \rangle = -k_1\langle N \rangle - k_2\langle N \rangle^2 - k_3\langle N \rangle^3,$$

where $\langle N \rangle$ is the average population of carriers per NC; k_1 is the trap-assisted recombination rate, k_2 bimolecular recombination coefficient; and k_3 is the Auger recombination coefficient. By global fitting this equation to our power-dependent kinetics with k_1 , k_2 and k_3 as the shared parameters, we could obtain values for these coefficients, i.e., $k_2 = 6.4 \times 10^{-5}$ (NC) ps^{-1} and $k_3 = 6.4 \times 10^{-5}$ (NC) 2 ps^{-1} . Meanwhile, the maximum range of our delay time measurement is too short for an accurate determination of k_1 , hence it is set to be zero. The result is shown in **Figure R2.4a**.

Figure R2.4 (a) Power dependent kinetics of MAPbI₃ colloidal NCs with 2.07 eV pump, for the determination of bimolecular and Auger recombination coefficient. (b) Pump-probe (PP) and pump-push probe (PPP) kinetics of our sample reported in the manuscript (i.e., 2.07 eV pump at 10 $\mu\text{J}/\text{cm}^2$ and 1.03 eV push at 1 mJ/cm^2), together with the simulated post-push dynamics from the obtained bimolecular and Auger recombination coefficients.

By using these values, we could simulate the corresponding carrier dynamics after-push due to the additional 20% carriers from the push ground-state absorption. The simulated results are reported in **Figure R2.4b**, where they are overlaid with our experimental data. The green-solid and red-dotted lines describe the simulated dynamics using the k_1 , k_2 and k_3 values for initial carrier populations of 100% (at 10 $\mu\text{J}/\text{cm}^2$ pump) and 120% (due to extra 20% from push ground-state absorption), respectively. It is worth to note that the additional carriers indeed cause slightly faster decay dynamics, but the estimated change is well within the experimental noise. Thus, we conclude that the Auger effect caused by 2PA of the IR push cannot account for the considerable lifetime variation observed. Therefore, a different kinetics model based on our original interpretation of hot carrier (HC) trapping is needed to fully describe the faster carrier recombination in the post-push dynamics. Herein, after the initial push fast relaxation (i.e. $t > 5$ ps), we proposed that a fraction of the carriers was ported over to the trap potential surface and undergoes an exponential decay via the trap-assisted recombination with a rate of $k_{\text{tr}} \sim 0.01 \text{ ps}^{-1}$ (as our original interpretation described in the Main Text); while the remaining fraction stays at the free-carrier potential and undergoes decay via the normal k_1 , k_2 , and k_3 rates. The resultant dynamics is shown as the solid red line in **Figure R2.3b**, where it successfully reproduces the experimental results.

Hence, we conclude that the reviewer has correctly suggested a role played by enhanced Auger recombination from the push ground-state excitation in causing faster recombination lifetimes. The Reviewer's interpretation is indeed confirmed in our visible push results, where there is a much higher ground-state absorption by the push and thus the interplays between these trapping and Auger dynamics (which we will discuss later in the comment #2 by Reviewer #1). However, in the case of IR push, this role of Auger is marginal and is not

sufficient to describe the lifetime variation that we observed. Thus, we believe that our interpretation of hot-carrier induced trapping as the dominant mechanism is still valid to complete the photophysical picture of the system.

- Page 11, Line 218: A discussion is added: *“In principle, enhanced Auger recombination from push-ground state absorption processes could cause an analogous spectroscopic response after the push. For instance, an IR-push (1.03 eV) could generate an additional population from the push ground-state absorption (via two-photon absorption processes), which would be invisible to our detection scheme. In this case, the additional population could enhance the bimolecular and trimolecular (Auger) recombination processes, thereby resulting in faster decay in our PPP kinetics. However, we quantified that this Auger contribution is minute for the case of our IR-push (Supplementary Note 1). On the contrary, we could demonstrate that the observed additional lifetime component arises from enhanced monomolecular recombinations (i.e., trapping).”*
- SI, Page S12: We replaced the extrapolated MAPbI₃ NCs absorption cross section at 600 nm with a directly measured one. The value is reported in Figure S15 and values of $\langle N \rangle$ are corrected accordingly throughout the Main Text and SI.
- SI Page S19: The results and considerations on artifacts caused by Auger effects, and on the opening of new trap channels, were added in the SI as part of the *Supplementary Note 1*.

2)The diagrams of potential energy surfaces in the model (Fig. 3) are problematic. It is hard to image that the PESs of hot and cold carriers are not crossed. In current pictures in Fig.3, most of hot carriers will transfer to the trapped carries, which certainly is not consistent with

the experimental result. I'd suggest that the authors carefully review the available knowledge about the band structure of the materials and make the model diagram more realistic.

We believe the Reviewer raises a good point. Our previous schematic used to describe the hot-carrier cooling was derived based on a previous publication (PRL110,067402 (2013)), where the hot-carriers were assumed to sit on a different PES with respect to the cold carriers. After careful consideration, we agree with the Reviewer that this description is not the most accurate one. Therefore, we redesigned our schematic according to the most recent reports [DOI: 10.1038/s41467-019-12830-1, and DOI: 10.1038/s41467-018-04946-7], which more correctly describe the hot carrier cooling on a single PES. Based on this feedback, Figure 3 in the main text was replaced by the following Figure and added the following reference for the band structure *J. Phys. Chem. C* **2015**, 119, 25209–25219.

- Page 14, Line 303: New Reference 56 was added in the revised version of the manuscript for the description of the perovskite electronic band structure.
- Page 15, Line 313: Figure 3 was replaced with a corrected representation of the free carriers and traps PES:

Revised Figure 3: ... (c) Vice versa, lower δ_E results in lower trapping rate and thus results in a lower number of trapped carriers, i.e., higher PLQY. *HC and CC indicate the hot carriers and cold carriers, respectively.*
Model fitting of the photo-action spectra ...

In my opinion, temperature-dependent measurements are highly preferred to justify the proposed model if applicable (optional).

We thank the Reviewer for the suggestion. Unfortunately, it has to be considered that low-temperature measurements can only be performed in the solid-state samples. While some types of solvents would allow low T measurements in solution samples (i.e., isopentane and methylcyclohexane mixtures), they are not suitable to synthesize stable PNCs dispersions. Although we agree with the Reviewer on the significance of temperature-dependent measurements, we also believe that performing these experiments in the solid-state would introduce more variables and processes to account for in the data analysis. Therefore, our choice of performing measurements on colloidal solutions restricted to room temperature helps to avoid the complications arising not only from possible phase transitions but also carrier diffusion and the consequent increase in the trapping rates, among others.

3) In addition to the PLQYs at different excitation photon energies, the spectra of PL excitation should be included which, in principle, can better reflect the excitation energy dependences of PL emission in the samples.

We thank the Reviewer for the comment. In the literature, photoaction spectra were reported both through direct PLQY measurements and with a relative method that considers the PL excitation profile and compares it to the absorbance spectrum [J. Phys. Chem. C 2011, 115, 45, 22089-22109]. Although both methods provide a representation of the photoaction spectra, we believe that the direct measurement of the PLQY is more reliable and less affected by scattering artifacts. Furthermore, the PLE/Abs method is based on two different

instruments, which would introduce more uncertainties. In Figure **R2.5**, we report the comparison between the two methods for the sample studied.

Figure R2.5 Comparison between photoaction spectra for MAPbI₃ (a) and MAPbBr₃ (b) colloidal NCs. Solid lines represent the normalized photoaction spectrum obtained as the ratio between photoluminescence excitation and absorption spectra. Wine and Blue squares represent the PLQY directly measured.

For both samples, the comparison between PLE and absorption follows closely the trend reported by the direct PLQY measurements. However, the former slightly overestimates the decrease in PLQY with respect to the direct PLQY measurement. We believe that this minor discrepancy could be ascribed to scattering artifacts. Indeed, scattering can increase the observed absorbance (i.e., extinction) and therefore result in lower values of the photoaction spectra.

- SI Page S4: These results were added in the SI as Figure S4
- Page 6, Line 120: These results are cited in the main text as: “*As reported in Figure S4, similar results are obtained by measuring the photoluminescence excitation spectrum and taking its ratio with the absorption spectrum.*”

Please also check the stoichiometric ratios of the samples studied. If miscellaneous phases and structures with higher bandgaps (PbI₂, or other low dimensional perovskites) exist in the samples, the lower PL QYs with higher excitations can be naturally explained.

Although we agree that the presence of different phases could affect the PLQY at different excitation energies, we believe that the combined absence of RP and PbI₂ reflections (<14°) in the XRD spectra (Figure S1) and the absence of additional optical edges in the absorption and transient absorption confirms their absence.

4)Optical stark effect is known to be important in these samples. Could the short-lived changes in Fig. 2b be relevant?

We thank the Reviewer for the suggestion. However, we can safely rule out this process, due to two main reasons: (i) The detuning energy ($\Delta = E_g - \hbar\omega_{\text{pump}}$) in our case is too large for the optical Stark effect to occur (i.e., $\Delta \sim 690$ meV). Based on the result reported by Beard et al. [*Nat Commun.* **7**, 12613 (2016)] in 3D MAPbI₃ perovskite, the optical Stark effect is only observed with very small detuning energy (i.e., $\Delta \sim 50$ meV). Since the effect is inversely proportional to Δ , we believe this is not the case. (ii) The optical Stark effect would result in a signal only during the pulse duration ~ 100 fs. However, in our case, our dynamics was proven to last longer than the pulse duration, i.e., ~ 500 fs. (3) Optical Stark effect would result in a differential like transient signal around the band-edge of the material, which is not the case in our study.

Moreover, some minor points should be carefully addressed.

5)Table 1, “Linewidth (eV)” seems to be “meV”? Figure S8 label for x-axis should be “ns”?

We thank the reviewer for the pointers, we have amended the typos.

6)Most of the traces are plotted in normalized scale. I'd suggest the authors to include the exact amplitudes of the signals for references by future studies in the community.

A similar comment was raised by the other reviewers, and our response is replicated here for convenience.

As mentioned above, the push beam is sequentially modulated on and off to obtain the PP and PPP, respectively. This allows us to avoid any possible instability in the laser power or changes in the overlap that can adversely affect the data. The differential transmission data are normalized to aid comparison. Data that is not normalized has been included in the revised SI, and it can be seen that the trends are identical.

Overall, this work is interesting and valuable for the community. I'd recommend publication of this manuscript if my major concern can be properly addressed.

We thank again the Reviewer for his/her constructive and thoughtful comments.

Reviewer #1

The manuscript by M. Righetto et al. discusses the details of hot carrier relaxation and the influence of the trapping centers on the photoluminescence quantum yield (QY) of MAPbX₃ perovskite nanocrystals. Authors employ steady state QY measurements and femtosecond pump-probe (PP) and pump-push-probe (PPP) spectroscopy as well as kinetic equation modeling to quantify the influence of excess excitation energy (δE) on QY.

The concept of the measurements and the results of this work are clearly presented. The manuscript reads well and discussion is presented without any hindrances/avoidances. Although the effect of excitation wavelength on the QY of nanocrystals, including perovskites, has been sporadically and intermittently discussed, this work presents the first comprehensive study that shows clear influence of δE on the emission properties of perovskite NCs. Thus, the manuscript presents an important advance in the field and will be relevant to the majority of researchers working in the field of perovskites. At the same time, I feel that several important questions need to be resolved before the manuscript could be considered further.

We thank Reviewer #1 for encouraging comments and constructive scientific feedback. We are greatly appreciative of all his/her inputs to further raise the standards of our manuscript to even higher levels.

1) Authors employ PP spectroscopy at different excess energies, which is not the easiest thing to do. Why not record time-resolved PL under the same conditions? Whenever the hot carriers experience the increased scatter/loss, it should result in faster recombination kinetics. I'm also concerned that using high fluences in push experiment (up to 1.7 mJ/cm²) may result in some unwanted artifacts and/or sample degradation. Using time resolved PL that allows many orders of magnitude lower pump fluences will be much safer in terms of sample degradation for which perovskites are well known.

We thank the reviewer for raising this point. We employed energy-resolved pump-probe (PP) using low pump fluences under three primary considerations: (i) PP spectroscopy and TRPL spectra will give equivalent information. Furthermore, PP yields better time resolution (ii) PP (using a white light probe, and therefore monitoring the transient absorption) probes the state-filling of both emissive and non-emissive states. Hence, it gives access to a broader range of states and not only the emissive ones that are only probed by TRPL. Therefore, checking the consistency of the spectrum allowed us to exclude the presence of other mechanisms. (iii) With our current experimental setup, the noise associated with the TRPL measurement at the same fluence is higher than the noise for the PP measurement.

Regarding the stability of the samples during the measurements, we performed all our non-linear experiments in cuvettes sealed under a nitrogen atmosphere and magnetic stirring. These experimental conditions allowed us to: (i) avoid possible degradation from photogeneration of reactive species, and (ii) providing a fresh sample at every pulse, thus drastically reducing possible thermal effects. To confirm this, we measured the linear absorption spectrum before and after a PPP measurement (repeated from Figure 2c). The substantial stability (**Figure R1.1**) throughout repeated measurements (up to 12 hrs) confirms that degradation processes throughout the measurement are negligible.

- SI Page S15: These results were added in the revised SI as Figure S18.

Figure R1.1 Evolution of the absorption spectrum of MAPbI₃ colloidal NCs during repeated PPP experiments. Samples were stirred and excited with the following PPP scheme: pump 2.07 eV (at 10 μJ/cm²), push 1.03 eV (at 1 mJ/cm²).

2) Discussion about PPP displayed in Figure 2(c,d) should be better articulated. It is not clear why the push at lower $\delta E = 1.03$ eV in panel (c) results in the appearance of a faster component in kinetics while at the same time, push with larger energy $\delta E = 2.07$ eV in panel (d) seems to provide same decay components.

We thank the reviewer for raising this point about the clarity in our interpretation. To summarize, the purpose of our PPP experiment with different push energies (Revised Figure 2c and Figure S4) is to expound our observation of excitation energy-resolved PLQY data, which show the presence of carrier losses at higher excess energies. Our PPP results show two different ultrafast push dynamics for IR push (sub-ps recovery) and visible push (no ultrafast recovery); while both push energies show faster recombination dynamics of the remaining carriers. The details of our interpretations of these observations are as the following:

1) For the IR energy (PPP with $\delta E = 1.03$ eV, Figure 2c), we observe the ultrafast rise of a positive $\Delta\Delta T$ signal (reduced bleaching) and its complete recovery with a time constant $\tau =$

500 ± 100 fs. Following this complete recovery, we observe the emergence of faster recombination rates ($k_r \sim 0.01 \text{ ps}^{-1}$) with respect to the simultaneously measured PP. We ascribe this effect to the effect of shallow trap potential populated by the IR push pulse (at the same level as the band-edge, and therefore still contribute to the bleaching signal), that can promote non-radiative recombinations. Thus, a full recovery of $\Delta T/T$ signal followed by a faster carrier recombination is observed. We describe this in the model presented in the paper, where the energy excess provided by the push promotes the overcoming of the trapping barrier. This mechanism is further confirmed by our kinetic simulations that include both Auger effects and hot carrier induced trapping processes, as suggested by Reviewer #2 (**Figure R2.4b**).

2) For higher push energies (PPP with $\delta E = 2.07 \text{ eV}$, Figure 2d), we observe the ultrafast rise of a positive $\Delta\Delta T$ signal (reduced bleaching) and its persistence (i.e., the absence of a thermalization/recovery). This is a signature of hot carrier trapping to high-lying traps, where carriers have sufficient energies to access these trap states upon push excitation – and thus not detected by our ground-state bleaching probe. Similar trapping processes under UV light illumination were reported by Klimov for CdSe QDs, and assigned to the trapping to high-lying defect states/ligand shell. Here, as pointed out by the reviewer, a faster recombination kinetic after the push is also observed. However, it is not expected since the pushed carriers are trapped at high lying trap levels and should not affect the remaining ground-state carriers. In our previous interpretation, we tentatively assigned it to “*the change in the trapping nature*”, without any further justification.

However, based on further experiments that we presented in our reply to Reviewer #2, we can now safely ascribe the presence of this faster recombination kinetic in the visible push

PPP experiment to the effect of ground-state absorption of the push. Similar to the case of IR push presented in our reply to Reviewer#2 (**Figure R2.3**), we estimated the maximum number of carriers excited by the ground state absorption of the visible push via fluence-dependent pump-probe experiments (**Figure R1.2**).

Figure R1.2 Pump-probe kinetics at the band-edge (1.72 eV) of our MAPbI₃ colloidal NCs, pumped with 2.07 eV at 10 μJ/cm² and 60 μJ/cm², respectively. Comparing the two populations, the visible push can generate up to 530% of carriers of those generated by the visible pump.

As shown in **Figure R1.2**, the ground state absorption of the visible push is indeed much higher than the initial excitation provided by the pump and can reach up to 530% of the initial value. Although our experimental configuration is not sensitive to the additional population arising from the push ground-state absorption (see comment #1b by Reviewer #2, page 3-4), the increased number of carriers in the excited state can still affect the subsequent kinetics. Using the recombination constants extracted from **Figure R2.4** (page 7), we could simulate the corresponding carrier dynamics after-push due to these additional 530% carriers from the push ground-state absorption. The result is reported by the dotted red line in **Figure R1.3** and

perfectly matches with the experimental data. Therefore, in the case of the visible push, we clarify that the faster recombination is originated by the enhanced Auger recombinations, arising from the visible push absorption from the ground state. However, having this said, we would also like to emphasize that the signal disappearance (i.e., the positive $\Delta\Delta T$ signal) is still likely to be caused by trapping of the hot carriers by high-lying traps, as in our original interpretation.

Figure R1.3 Pump-probe (PP) and pump-push probe (PPP) kinetics of our sample reported in the manuscript (i.e. 2.07 eV pump at 10 $\mu\text{J}/\text{cm}^2$ and 2.07 eV push at 60 $\mu\text{J}/\text{cm}^2$), together with the simulated post-push dynamics from the obtained bimolecular and Auger recombination coefficients.

It is worth to note that after much re-consideration, we decided to relegate the original Figure 2d (kinetics by visible push) into the Supporting Information. Our rationale is that this paper focuses mainly on the extent of the perovskite defect tolerance in the presence of hot carriers. Results from the IR push experiment suggest that trapping is enhanced upon push-induced heating and therefore call shallow traps into play. Our model formulated based on this interpretation of shallow trap potential could also consistently describe all our experimental

observations. On the other hand, our vis-push (2.07 eV) PPP experiment suggest the presence of higher-lying traps, which, while might have some implication for UV excitation with $\delta E > 2$ eV, it is beyond the scope of this study.

- Page 9: We moved Figure 2d to the Supporting Info as Figure S9 of the revised SI
- Page 11, Lines 210-216: We replaced the visible push interpretation paragraph with a discussion of possible results and artifacts. Thus, we removed “*On the other hand, the hot carrier behavior when pushed at 2.07 eV ($60 \mu\text{J cm}^{-2}$) is entirely different (Figure 2d), thus implying that push energy has a significant effect on the subsequent carrier dynamics. The different behavior indicates the activation of a different mechanism: under these conditions, the thermalization process is absent, and no contribution to the photobleaching is preserved. This suggests that after pushing the carriers, they are either trapped in higher energy states or transferred out of the NCs.⁴⁰ Moreover, an additional fast decay component is also observed in the PPP ΔT signal, similar to when pushed at 1.03 eV. Therefore, our results suggest that even higher energy pushes affect the nature of the trapping and induce faster recombination when they relax back to a cold state, thereby suggesting that two different mechanisms exist within the ensemble population.*”, and replaced it with “*The hot carrier behavior for a high energy push (2.07 eV, $60 \mu\text{J cm}^{-2}$) is entirely different and indicates a sudden loss of carriers when the push is absorbed (Figure S9). This sudden disappearance of the carriers could indicate that at higher excess energies other carrier losses mechanisms are present (i.e. higher-lying traps), analogous to previously observed photoionization mechanisms.⁴⁴ Moreover, a faster recombination dynamics after the push is also observed for the visible-push.*”
- Page 12, Lines 259: We added the paragraph from: “*On the other hand, the presence of push ground-state absorption is more significant when employing higher energy pushes*

(i.e., our visible 2.07 eV push), due to direct competition between the ground-state and the excited-state absorption processes. In this case, our analysis in Supplementary Note 1 shows that enhanced Auger dominates the recombination mechanism after the push. Hence, although the observed positive ΔT upon visible push absorption is a strong indication of possible trapping at high-lying trap sites, the presence of overlapping spectroscopic responses makes challenging their complete disentanglement. Therefore, we limit our further investigation to lower energy excesses.”

- Page 12, Lines 259: We rephrased the paragraph from: “These observations suggest the *presence of* shallow trapped carrier states, where one carrier (either the electron or the hole) gets trapped at a shallow trap site and the other still contributes to the bleaching *while displaying stronger Coulomb interactions and therefore faster recombination.*” to “These observations suggest *that the hot carriers could promote the formation of* shallow trapped carrier states, where one carrier (either the electron or the hole) gets trapped at a shallow trap site and the other still contributes to the bleaching. *This would explain the observed faster recombination after the push as increased non-radiative recombination caused by shallow trap states.*^{45,46} *Our results suggest that this trapping takes place on the same timescale as the cooling. Analogous considerations on HC trapping were reported in past studies on CdSe⁴⁷ and recent studies on MAPbI₃ thin films.¹⁹”*
- SI, Page S19: We summarized all the discussion on the effect of the enhanced Auger Recombination in the Supplementary Note 1.

I would also suggest having dynamics without normalization because it is unclear how much signal is lost from the regenerated hot carriers when they get trapped.

The differential transmission data are normalized to aid comparison. Data that is not normalized has been included in the revised SI, and it can be seen that the trends are identical.

- SI, Page S15: We added the raw data (not normalized) for comparison, in Figures S19-S23.

3) For MAPbI₃ NCs, authors reach $\delta E > 2$ eV, which is more than twice the bandgap energy. Recent work has shown carrier multiplication (CM) in CsPbI₃ NCs, (de Weerd et al., Nat. Comm. (2018) 9:4199 | DOI: 10.1038/s41467-018-06721-0) at such excess energy. Authors should corroborate on how their results might relate and could CM contribute to the drop in PLQY at higher energies.

We thank the reviewer for this interesting observation. Following the additional results presented in comment #2, we decided to limit our considerations to lower push energies, due to the presence of overlapping spectroscopic responses when visible pushes are employed. However, the Reviewer poses an interesting question that mostly concerns the nature of the push induced signal. Indeed, the combination of band edge carrier energy (1.7 eV) and push energy (2.07 eV) exceeds the $2E_g$ threshold. In principle, this would allow the presence of carrier multiplication (CM). However, we believe that the observed dynamics does not depend on CM effects because of two main considerations:

- 1) CM would determine an increase in the number of band-edge carriers and therefore would result in a negative $\Delta\Delta T$ signal. On the contrary, we observed a reduction in the number of carriers, which suggests that carrier losses mechanisms in action, rather than carrier multiplication mechanisms.
- 2) In our recent publication, we observed similar behavior for high energy pushes also for bulk MAPbI₃ (Lim et al., Sci. Adv. 2019; 5:eaax3620). Since CM is strongly dependent

on the dimensionality and not expected to occur in the bulk due to momentum conservation considerations, we believe CM plays a negligible role in the observed dynamics.

Thus, we conclude that in the case of our MAPbI₃ NCs samples, PPP allows us to observe trapping and not CM. Herein, it is worth to note that these samples differ in size, shape, and composition from the samples used to demonstrate CM. Therefore, we believe that this very interesting comment opens to the question of whether we can use PPP to monitor CM, and we foresee that this could be the subject of a future study.

4) Authors develop modeling based on Marcus theory with corresponding rate equations. As it looks, it is entirely phenomenological model. It fits the observed kinetics based on rate constants, reorganization energy and coupling energy. However, as any other such model, it cannot predict the nature of the trapping centers. I think it is very important to develop first-principle model that can do that. As an example, it has been established via theoretical modeling that Br-vacancy centers are responsible for changes in PL in CsPbBr nanoparticles. As a result, high PLQY samples have been developed. Given a sufficiently large range of parameters in the phenomenological models, it is often possible to provide fitting to nearly every observable kinetic by simply adding more free parameters, which may or may not be of physical sense.

We thank the Reviewer for the comment. However, we respectfully disagree with the Reviewer for four main reasons. Firstly, we would like to clarify that all fitting parameters used in the model have their physical significance in the dynamics. Each term in the coupled rate equations (i.e., Equation 1 and 2 in the Main Text), describes an important part of the dynamics and has already been described as the trapping/detrapping model [*J. Phys. Chem. Lett.* 2018, 9, 4955–496, and *PNAS* 106, 9, 3011 (2009) and *Phys. Rev. B* 87, 081201 (2013)]

]. Secondly, we used a restricted range of free parameters in our fitting (i.e., the electronic coupling constant H_S , the reorganization energy λ , and the bimolecular recombination rate, k_2), while the rest of the parameters are determined phenomenologically from the results of other experiments. Thirdly, our fitting produces physically reasonable values of λ (i.e., make a good physical sense and is comparable with reports from another QD system [*PNAS* 106, 9, 3011 (2009) and *Phys. Rev. B* 87, 081201 (2013)]). Lastly, from our PLQY model fitting, we obtained $k_2 \approx 10^{-7} \text{ cm}^3\text{s}^{-1}$ as value for the bimolecular recombination rate constant. Considering the concentration of our NCs samples to be $\sim 2 \times 10^{14} \text{ NC/cm}^3$, this k_2 value corresponds to $k_2 = 2 \times 10^7 \text{ (NC)/s} = 2 \times 10^{-5} \text{ (NC)/ps}$. This simple estimate is consistent, within the order of magnitude, with the value of $k_2 \sim 6 \times 10^{-5} \text{ (NC)/ps}$ obtained from independent power-dependent pump-probe measurements performed for this response letter (**Figure R2.4**). Therefore, these reasons give us more confidence in the accuracy of the model.

I think that absence of the first-principle calculations of defect formation is the main weakness of this work and should be addressed before the publication is possible. The authors themselves do experiments with TOPO ligands, which is a common substance for NC passivation and observe modifications. However, without the predictive model, further ligand modification seems to be difficult and would simply result in a trial and error approach.

The Reviewer raises an important point concerning the role of first-principle calculations to substantiate our findings. In principle, we would agree with the Reviewer, first-principle calculations for defects are a powerful tool to increase the predictive impact of our findings. However, the nanometer size of our samples largely limits the effectiveness of theoretical calculations. Indeed, to the best of our knowledge, the paper by Infante *et al.* [*ACS Energy Letters* 2019, 4, 11, 2739-2747] represents the state-of-the-art of calculation on defects in

nanocrystals. In this paper, calculations were limited to 2.9 nm-sized CsPbBr₃ clusters, without the possibility of including spin-orbit coupling. This limitation makes clear that, even though first-principle calculations are a powerful tool for the advancement of nanocrystal science, the computational effort and resources behind these calculations is extremely large and it is beyond our current capabilities.

On the contrary, defects in bulk lead halide perovskites were extensively studied in the past years. The rationale behind our study builds on the well-established knowledge of defect formation energies in bulk. Our PPP spectroscopy results identify shallow traps as possible sources of increased non-radiative recombination rates, activated by carriers with excess energy. As described in the main text, from References 6,7 halide vacancies (V_X^+) are abundant (low defect formation energy) and introduce shallow traps. Therefore, we respectfully disagree with the Reviewer: we based our approach on previous knowledge of the perovskite surface chemistry. Our paper aims to demonstrate the presence and the spectroscopic signatures of hot carrier trapping in perovskite nanocrystals. Our experiments with TOPO served a two-fold purpose of i) confirming that what we observe is related to surface defects and not to other dark states: ii) indicating a future direction for mitigating this issue. Hence, we do agree that the suggested approach would be valuable, but we believe that it should be itself the subject of a future study.

- Page 17, Line 354: We added a brief phrase to clarify the role of our ligand exchange experiments: “Here, the comparison between differential PPP kinetics for pristine and ligand-exchanged MAPbI₃ evidences how the additional TOPO passivation results in lower carrier losses to both ligands and shallow traps sites. *Therefore, these experiments confirm that the observed reduction of the PLQY related to the surface passivation of the perovskite NCs. Specifically, we observe that traps sites that are supposedly benign or*

“inactive” for the cold carriers,⁶ and whose passivation is generally not considered as necessary, become activated under the energy (or the “heat”) of the hot carriers.

5) There is a confusion about the first term in Eq 1. It shows N^2 which is true for radiative recombination involving free carriers. In the text authors mostly refer to excitonic behavior and it makes things confusing. Only reading Suppl. Info it becomes more clear that for different samples (iodide vs. bromide) different terms are used. Please clarify it in the main text.

We thank the reviewer for helping us to improve the readability of our manuscript. We made the description of our model more clear in the main text. Adding the following phrases:

- Page 13, Line 282: We added a comment that explains the use of different models for iodide and bromide-based perovskite nanocrystals carrier recombination: *“In agreement with Elliott fitting results, the carrier population term presented in Eq. (2) and Eq. (3) is modified in MAPbBr₃ NCs modelling, to account for the more excitonic nature of the recombination these samples. Specifically, the radiative bimolecular recombination term is replaced by a monomolecular radiative recombination term with rate k_r .”*

We are greatly appreciative of all the Reviewer’s inputs that further raised the standards of our manuscript to even higher levels.

Reviewer #3

The manuscript by Righetto and coworkers reports a spectroscopic study of hot carrier trapping in MAPbI₃ and MAPbBr₃ perovskite nanocrystals using a combination of "pump-push-probe" spectroscopy and PLQY measurements. The results are rationalized in the framework of the Marcus charge transfer model.

I believe the overall topic and approaches in the paper are interesting to the perovskite community, but I have several concerns which I feel should be addressed before it can be published in Nature Communications.

We thank the reviewer for the encouraging opinion and constructive comments to our manuscript.

1. There is very little discussion about previous works on the role of defects and overall material quality on hot carriers in perovskites. How do the authors reconcile their observations with previous reports, especially those that indicate the absence of hot carrier trapping (i.e. J. Phys. Chem. C 2017, 121, 21, 11201-11206, DOI: 10.1021/acs.jpcc.7b03992)?

We thank the reviewer for raising this point. The literature on the carrier traps in perovskites is broad and reports sometimes contrasting results. For instance, other very recent reports by Harel et al. [*ACS Energy Lett.* **2019**, 4, 1741–1747], Samanta et al. [*ACS Nano* **2019**, 13, 11, 13537-13544], and Kim et al. [*Small* **2019**, 15, 1900355] suggest that hot carriers are influenced by traps, as opposed to the references suggested by the Reviewer. Therefore we believe this topic is currently still an open question, which we seek to clarify from this study. To elaborate, we believe that this variability arises in part from the difference in the sample studied. Moreover, the work cited by the Reviewer reports very long hot carrier lifetimes (40 ps). It is commonly accepted that these HC lifetimes for MAPbI₃ can be achieved only in the

“Auger reheating” regime. Therefore, we believe that the absence of hot carrier traps could possibly be ascribed to trap flooding at higher carrier injections.

- Page 4, Line 59: We added a comment on the previous studies in the Introduction section, citing the relevant literature: “*Few recent works 17,18 reported contrasting results on the presence of this coupling, thereby leaving an open question.*”

2. Can the authors comment more on (i) the timescale for (hot) carrier trapping, (ii) the energy of the trap states in the studied systems and (iii) whether hot and cold carriers are affected by the same traps?

The Reviewer raised an interesting point. Our results indicate that, when low excess energy are provided (i.e., by our IR push), excess energy promotes the trapping of the carriers to band-edge shallow traps. This hot carrier (HC) induced trapping occurs within the same timescale as the HC cooling or thermalization time, as have been shown in our pump-push-probe experiment and as previously suggested for CdSe QDs by Kambhampati [*J. Chem. Phys.* 129, 084701 (2008)]. In our 1.03 eV push experiment, it has been shown that the HC relaxed to the trap potential (i.e., being trapped), within the relaxation time of ~500 fs. In a very recent report on bulk MAPbI₃ perovskite Harel [*ACS Energy Lett.* **2019**, 4, 1741–1747] has reported an only slightly faster HC cooling at trap sites, therefore we can speculate that what we observed is an average between trapping and cooling. On the other hand, when high excitation energies excesses are provided, (e.g., by our visible push), we propose that carriers are trapped at higher-lying traps, such as ligand/surface states, in a mechanism that resembles that of photoionization.

- Page 12, Line 258: A short comment highlighting the hot carrier trapping timescale when IR excitation and relevant citations are added: “*Our results suggest that the trapping takes*

place on the same timescale as the cooling. Analogous considerations on HC trapping were reported in past studies on CdSe⁴⁷ and recent studies on MAPbI₃ thin films.¹⁸”

(ii) the energy of the trap states in the studied systems

As discussed in the comment #4(b) by Reviewer 1, the observed trap potential is likely due to halide vacancy traps, whose energy level is located coincidentally within the conduction band-edge of the system [by DFT calculations (DOI: 10.1039/C9MH00500E)]. This assignment is also consistent with both our PPP results and the passivation by TOPO. In our PPP result by IR push, we observed full recovery of the initial bleaching signal after the push, yet faster recombination afterwards due to trapping. We believe this is due to the trapped carriers are still contributing to the bleaching signal and the trapping is at about the same energy level as the band-edge. Our rational choice of TOPO passivation is also aimed to passivate the under-coordinated lead ions as a result of halide vacancies. Since this passivation positively affects the PLQY values, we believe this further confirms our hypothesis.

and (iii) whether hot and cold carriers are affected by the same traps?

The Reviewer’s question regarding the energy of the traps is an interesting and important one. Although our phenomenological model cannot provide accurate values on the energy of these traps, our work gives some indications on the types of traps playing a role in different excitation conditions. Based on our results we can infer that there are at least three different kinds of traps:

1. Surface/Ligands/higher-lying traps removing HC population, can be accessed with a visible push.

2. Shallow Traps that requires carrier thermalization and contribute to the PP signal through stimulated emission signal, observed in the IR push effect.
3. Deep Traps, whose presence can be inferred by PLQY values <1 even for band edge excitations.

Hence, our results indicate that while cold carriers are mainly affected by deep traps, increasing amounts of excitation energy excesses promote the population of shallow traps. When the excitation energy is consistent (visible push), other HC traps come into play.

3. Compared to 1-sun illumination, transient optical experiments are performed at relatively high intensities. Can the authors comment on the significance of hot carrier trapping under the standard operational conditions of a device?

We thank the reviewer for raising this critical point. We do agree on the fact that the fluence used in transient optical experiments would yield results that cannot be compared with 1-sun illumination, in terms of carrier injection. However, since we are studying traps we believe that using fluences low enough not to induce trap saturation in the perovskite material is the most correct approach. As demonstrated in our previous work [Supporting Info, Xing et al. Nat. Mater. 13,476–480 (2014)], the PLQY in MAPbI₃ thin films measured under 600 nm excitation starts saturating above 25 μJ/cm². Therefore, we believe our approach using a conservative 10 μJ/cm² excitation allows us to observe the trapping, especially considering that in PPP experiments only a fraction (~30%) of the carriers are pushed to higher energy states. Lastly, we believe that the significance of hot carrier induced trapping in standard operational conditions is further confirmed by PLQY measurements (indicating carrier losses) that were conducted at lower fluences (lamp excitation).

4. The manuscript focuses more heavily on the iodide systems. The general consensus in the literature is that the iodide-based perovskites are more unstable (less defect tolerant) than the bromide equivalents. Based on the data in the manuscript, is hot-carrier trapping more prevalent in MAPbI₃ NCs than MAPbBr₃ NCs? For instance, the bromide samples exhibit a higher overall PLQY (Figure 1a+b) and a lack of significant push-induced changes in Figure S9.

We thank the Reviewer for the comment. Unfortunately, we cannot safely support the Reviewer's conclusion on the significance of hot carrier traps in different materials since it is drawn from PPP data. The reviewer correctly observes the difference in the PPP responses for iodide and bromide based perovskite nanocrystals. However, as we explained in the Main Text (Page 12 Line 242), we ascribe this difference to the more pronounced excitonic character in MAPbBr₃ NCs. In these excitonic systems, the high energy excitation produces a hot electron-hole pair that cools and forms cold excitons. Although during this cooling the carrier configuration is similar to that of MAPbI₃, the PPP operates on the re-excitation of these carriers. Unfortunately, the stronger excitonic coupling in MAPbBr₃ removes the presence of free carrier absorptions (usually falling in the NIR) and renders more difficult the re-excitation of the carriers. The need for higher pump and push fluences (40 $\mu\text{J}/\text{cm}^2$; 2 mJ/cm^2 , respectively) to observe a PPP response, and the derivative-like shape of this response further confirm our hypothesis, possibly calling into play the excitonic effect and optical Stark effect.

5. My most serious concerns are about the execution and interpretation of the pump-push-probe experiments. a) In the methods, the authors should explain how the 3.1 eV pump is generated for the experiments on the MAPbBr₃ NCs.

The 3.1 eV (400 nm) pump is generated using a BBO crystal placed in the path of the Ti-Sapphire laser's 800 nm fundamental output. We apologized if we have failed to mention this. We have therefore made the following revision:

- Page 21, Line 461: We explained the procedure in the Experimental Section: *“In the experiments on MAPbBr₃ NCs, the 3.1 eV pump is generated by the frequency-doubling of the residual fundamental output using a BBO crystal.”*

b) Can the authors provide more explanation for the differences in Figure S5? Namely, (i) could the different timescales be related to residual hot carriers and phonons from the initial pump? This might have implications for other data in the manuscript. (ii) What happens to the late-time dynamics when the push delay is increased?

The Reviewer raised an excellent point. In Figure S5, we observed a consistent minute decrease in the relaxation time with the increasing push delay. While this might be due to a deeper physics (e.g., the recovery from pump induced phonon bottleneck or the residual hot carriers as suggested by the Reviewer), our temporal resolution of ~100 fs prevented us from drawing a definitive conclusion on the observation. However, in any case, such processes fall outside of the scope of this paper. Indeed, we observe a similar behavior of the transients at longer time delays for different push delays, as reported in the revised SI (Figure S21). Faster recombinations after the push pulse are observed independently of the push delay. Hence, we conclude that these early time effects do not affect our interpretation.

- SI, Page S17: We added the complete transients for push delay dependent measurements in Figure S21 of the revised SI.

c) The PP vs PPP differential transmission data are all normalized, and there is no information regarding the modulation of the push beam. This calls into question the interpretation of the push-induced signals.

A similar comment was raised by the other reviewers, and our response is replicated here for convenience.

The push beam is sequentially modulated on and off to obtain the PP and PPP, respectively. This allows us to avoid any instability in the laser power or changes in the overlap that can adversely affect the data. The differential transmission data are normalized to aid comparison by simply dividing PPP and PP kinetics by the same constant. Data that is not normalized has now been included in the revised SI, and it can be seen that the trends are identical.

d) The push pulses are quite intense (~mJ cm⁻²). Do the authors observe any sample degradation (difficult to see with normalized data) under these conditions, and could this influence the overall carrier dynamics?

We thank the Reviewer for the comment. A similar point was raised by another reviewer and the reply is partially reproduced here. We performed all our non-linear experiments in cuvettes sealed under a nitrogen atmosphere and magnetic stirring. These experimental conditions allowed us to: (i) avoid possible degradation from photogeneration of reactive species, and (ii) providing a fresh sample at every pulse, thus drastically reducing possible thermal effects. To confirm this, we measured the linear absorption spectrum before and after a PPP measurement (repeated from Figure 2c). The substantial stability (**Figure R3.1**) throughout repeated measurements (up to 12 hrs) confirms that degradation processes throughout the measurement is negligible.

Figure R3.1 Evolution of the absorption spectrum of MAPbI₃ colloidal NCs during repeated PPP experiments. Samples were stirred and excited with the following PPP scheme: pump 2.07 eV (at 10 μJ/cm²), push 1.03 eV (at 1 mJ/cm²).

- SI Page, S15: These results were added in the revised SI as Figure S18

e) The "Push Delta" signals in Figure 2c+d look very different (the latter exhibits no ultrafast decay (carrier thermalization) after the push event). When pushing these MAPbI₃ NCs (1.73 eV bandgap) at such a high energy (2.07 eV), how can the authors possibly distinguish between the contributions from (i) the intended "heating"; of the excited (band-edge or trapped) states (ii) excitation into some higher energy (possibly "dark") electronic band, and most importantly (iii) excitation from the ground state? f) As noted in (d), the push intensities are quite high. Following from point (e), in the case of the sub-gap push energies, is it possible that multi-photon absorption (as in their own work from ref. 22) can contribute to the optical response in PPP?

The Reviewer raised a valid point. We agree with the Reviewer that the signal from the visible push (2.07 eV) could contain contributions from multiple sources, which could result in such different after-push dynamics. In fact, we assigned the “missing” carriers after the

600 nm push to the trapping by higher (possibly dark) energy states, which most likely comes from the ligands or other high-lying trap states, (i.e., process (ii), as mentioned by the Reviewer). Regarding the process (iii), our PPP measurement is insensitive to the push excitation from the ground state, due to our experimental setup that uses a chopped pump. Nevertheless, as a secondary effect, the push induced additional population could cause a slight variation in the subsequent dynamics both in the cases of visible and IR pushes. The response to this comment is therefore similar to the combination of our responses to Comment 1 raised by Reviewer 2 and Comment 2 raised by Reviewer 1 (Pages 3-8 and 17-19). Due to the length of our responses and for brevity, it will be only summarized here. Please refer to our complete response on pages 5-7 and 14-17.

Briefly, to quantify the maximum amount of additional carrier excited by the push, we performed “push-probe” experiments using both IR (1.03 eV) and visible (2.07 eV) pushes and replicating the fluences used in the PPP experiments (see **Figures R2.3** and **R1.2**, respectively). Results indicate that under IR push excitation up to 20% of the originally pumped carriers can be generated due to the two-photon absorption process, while under visible push the additional carriers can reach up to 530% due to the dominant linear absorption.

As mentioned above, we cannot observe this additional population using a chopped pump detection scheme but this population could influence the after-push carrier recombination dynamics. To quantify this, we estimated the carrier recombination constants in our NCs and simulated the recombination dynamics after the push considering the overall number of carriers. As shown in **Figure R3.2** (replicated from Figure R2.4b and R1.3), the additional carriers cause a negligible effect in the case of the IR push. In this case, the after-push carrier

recombination can be well explained by an additional trap term in the recombination model. On the other hand, in the case of a visible push, the effects of the additional population originated by the push ground state absorption on the subsequent dynamics are substantial.

Figure R3.2 (a) Pump-probe (PP) and pump-push probe (PPP) kinetics of MAPbI₃ NCs reported in the manuscript (i.e. 2.07 eV pump at 10 μJ/cm² and 1.03 eV push at 1 mJ/cm²), together with the simulated post-push dynamics from the obtained bimolecular and Auger recombination coefficients. (b) Pump-probe (PP) and pump-push probe (PPP) kinetics of our sample reported in the manuscript (i.e. 2.07 eV pump at 10 μJ/cm² and 2.07 eV push at 50 μJ/cm²), together with the simulated post-push dynamics from the obtained bimolecular and Auger recombination coefficients.

To help address points c-f, I recommend that the authors perform "push-probe"; control experiments (like pump-probe, but using a chopped push as the excitation beam) and compare this data with the PPP responses. To conclude, I think the manuscript would substantially benefit from additional discussions and relatively simple control experiments outlined above.

We are grateful to the Reviewer for providing us this scientific feedback. We believe that the control experiments reveal the Reviewer's comments were correct: there is a contribution to the optical response from the linear absorption (for the visible push) and a much smaller one from two-photon absorption (for the IR push). We have indeed performed the experiments

(**Figures R2.3 and R1.2**). However, we believe that our additional findings clarify that our overall interpretation is not affected. In the case of the IR push, we found that the additional optical response is marginal and cannot account for the observed dynamics after the push. On the contrary, this can be well simulated by including an additional trap term, thereby further confirming our model. Meanwhile, in the case of the visible push, we found that the observed faster recombinations are caused by the additional population induced by the ground state absorption of the push. However, since our technique is not sensitive to this population, we can confirm that our hypothesis of population loss occurring as a result of the push absorption is still valid. However, considering that the rest of the paper focuses on the shallow traps (i.e. the results from NIR push), we moved the report on the vis-push to the SI.

- SI, Page S19 : These considerations were added in the Supplementary Note 1

On a lesser note, there are some grammatical errors throughout the manuscript, but particularly in the abstract and introduction. For instance "its"; as opposed to "their" defect tolerance in the first line of the abstract, and "consist of predominantly of shallow traps"; on page 3

We apologize for the typos in the previous version. The current version has been proof-read by several native speaker authors and the errors have been amended.

Overall, I do not believe that the manuscript, in its current form, meets the standard of Nature Communications, and would recommend seeking publication in a more specialized journal such as JPCL.

We would like to thank again the reviewer for providing precious scientific feedback that further raised the quality of our work. We believe that this version of our manuscript is significantly improved and meets the high standard of Nature Communications.

Reviewers' comments:

Reviewer #1 (Remarks to the Author):

The authors made a very thorough work addressing mine and other reviewers points. I'm mostly satisfied with the answers and recommend the paper for publication.

Reviewer #2 (Remarks to the Author):

The authors have carefully addressed my concerns.

Reviewer #3 (Remarks to the Author):

I commend the authors for their efforts in carefully responding to the remarks of all three reviewers. As I stated in my previous review, I believe the overall topic and approaches in the paper are interesting to the perovskite community. Having said that, I still have a few comments and questions regarding the interpretation and communication of the experimental data.

1. I think that the authors could make a much stronger case for their story, and in responding to my previous queries, by directly comparing the recovery dynamics of the push-induced TA signal for the passivated and unpassivated NCs. This would:

- i. Reveal whether the intraband cooling dynamics are actually affected by the (higher-lying) surface traps or not;
- ii. Possibly provide some means of quantitatively assessing the timescale of hot carrier trapping with respect to cooling;
- iii. Possibly demonstrate whether hot and cold carriers are affected by the same traps (for instance if the ~ns carrier lifetime and cooling dynamics are both influenced by passivation)

On close inspection of Figures S8 and S17, I do not see much difference in the aforementioned recovery timescales of the pristine and w/TOPO sample, but showing the PPP or push delta traces for these samples at each given push fluence on the same graph would be helpful.

2. I see the authors have recently published work using the same pump-push-probe experiment on bulk MAPbI₃ (ref. 40 in the manuscript). Out of curiosity, how do the push-induced TA signals for bulk MAPbI₃ compare with the analogous NCs? This comparison may also help to shed light on the impact of the NC surface on cooling.

3. Please label the data in all the figures as "MAPbI₃ NCs" or "MAPbBr₃ NCs" (not just MAPbI and MAPbBr) to avoid confusion with the bulk materials.

4. If the results of this work are to be believed, it appears that the consequences of hot carrier trapping are mostly manifested at "late" times, i.e. after cooling has occurred. In a hot-carrier solar cell, extraction should outpace cooling. Could the authors provide some discussion connecting their results with this practical consideration?

5. Can the authors comment on why, in Figure S11(a), the PPP trace goes below zero norm $\Delta T/T$ at 3

ps?

6. Reviewer #2 raised an interesting point about the Stark effect. The authors stated in their response that this would lead to a “differential-like transient signal around the band-edge of the material”, but claimed that it is not significant in the presented data. How do the authors explain the trend in Figure S14(b), which shows a clear derivative-like signal, redshifted with respect to the bandgap of the MAPbBr₃ NCs? Previous pump-push-probe studies by the Friend group on organic semiconductors have attributed these effects to Stark shifting.

On a non-scientific note, I appreciate that the authors may not be native speakers, but I still have gripes with the poor grammar in the manuscript. I have identified the following recurring issues, which I hope the authors will find constructive for any future iterations of their paper:

i) General sentence structure. Example: on line 230, “the presence of overlapping spectroscopic responses makes challenging their complete disentanglement...” should read “the presence of overlapping spectroscopic responses makes their complete disentanglement challenging...”)

ii) Tense of clauses. In particular, the mixing of past and present tense within the same sentence. Example: on line 13, “Amongst the many spectacular properties of the hybrid lead halide perovskites, their defect tolerance is revered as the key enabler for a spectrum of high-performance optoelectronic devices” is in the present tense, but the remainder of the sentence on line 15 “that propelled perovskites to stardom” is in the past tense.

iii) Incorrect use of singular and plural terms. Example: on line 62, “Perovskite nanocrystals (PNCs) provides an exciting platform...” should read “Perovskite nanocrystals (PNCs) provide an exciting platform...”

In my opinion, the current manuscript is difficult to read, and the basic language issues above need to be addressed to meet the standards of Nature Communications.

Response to the reviewers

Reviewer #1

The authors made a very thorough work addressing mine and other reviewers points. I'm mostly satisfied with the answers and recommend the paper for publication.

Reviewer #2

The authors have carefully addressed my concerns.

We are delighted by the encouraging comments and constructive scientific feedback of Reviewer #1 and Reviewer #2. We believe that our paper has been further improved thanks to all their thoughtful inputs and comments.

Reviewer #3

I commend the authors for their efforts in carefully responding to the remarks of all three reviewers. As I stated in my previous review, I believe the overall topic and approaches in the paper are interesting to the perovskite community. Having said that, I still have a few comments and questions regarding the interpretation and communication of the experimental data.

We are grateful for these positive and constructive comments by Reviewer #3. Also, we are greatly appreciative of his/her inputs that proved very useful in the previous round of comments.

1. I think that the authors could make a much stronger case for their story, and in responding to my previous queries, by directly comparing the recovery dynamics of the push-induced TA signal for the passivated and unpassivated NCs. On close inspection of Figures S8 and S17, I do not see much difference in the aforementioned recovery timescales of the pristine and w/TOPO sample, but showing the PPP or push delta traces for these samples at each given push fluence on the same graph would be helpful.

We thank the Reviewer for this interesting comment. We do agree with the Reviewer: the comparison between time-resolved spectroscopy data for ligand exchanged and pristine nanocrystals is an interesting and informative one. Indeed, we have already addressed this matter briefly in Figure S16, by reporting the "fraction" of carrier losses for pristine and ligand-exchanged MAPbI₃ NCs. To elaborate on this point and answer to the questions, we present a more detailed comparison.

Figure R.3.1 Comparison between differential pump-push-probe signals for pristine MAPbI₃ NCs (blue) and TOPO-ligand exchanged MAPbI₃ NCs (red). Samples were pumped at 2.07 eV (37 μJ/cm²) and pushed with 1.03 eV pulses (1 mJ/cm²). The data shows a similar cooling negative ΔT/T signal at 1.07 eV, which implies the presence of excited-state absorption.

In **Figure R.3.1**, we show the direct comparison between the differential pump-push-probe signals for pristine and ligand exchanged MAPbI₃ NCs, previously reported in Figure S7 and S16. Quantitatively, the model fit reveals a similar cooling time (within our temporal resolution) of $\tau_{\text{thermal}} = 460 \pm 100$ fs for both pristine and ligand-exchanged MAPbI₃ NCs. However, the passivation substantially impacts on the long-time dynamics rather than the cooling dynamics. Our results show a decrease in the contribution of trap-assisted recombination by ~26% upon TOPO ligand exchange (i.e., the fractional decrease in the fitted lifetime component). This implies that such passivation reduces the number of trapped carriers among those re-heated by the push.

i. Reveal whether the intraband cooling dynamics are actually affected by the (higher-lying) surface traps or not;

The Reviewer posed an interesting question on the effect of the traps on the cooling times. Within our limited time resolution, we did not see any considerable change in the intraband cooling time. This is in line with a very recent report for bulk MAPI, where it has been shown that surface traps in grain-boundaries do not significantly affect carrier the sub-ps cooling rates (*i.e.*, Fig. 3e in Ref. 19). As a commentary on this, the cooling rate reflects the rate of energy loss by LO phonons in the system, which is strongly determined by the phonon density of states (DOS). From this result, we could infer a minimal effect of these traps on the phonon DOS.

ii. Possibly provide some means of quantitatively assessing the timescale of hot carrier trapping with respect to cooling;

Based on our findings, the best conclusion that we can draw is that the trapping to the shallow-trap potential and carrier cooling process are competing over a similar timescale. At this stage, given with our limited temporal resolution, we believe that a quantitative assessment of this timescale beyond this level would be an overinterpretation of the data.

iii. Possibly demonstrate whether hot and cold carriers are affected by the same traps (for instance if the ~ns carrier lifetime and cooling dynamics are both influenced by passivation)

Based on the results reported in **Figure R.3.1**, we can experimentally confirm that the additional passivation influences mainly the $\sim 10^2$ ps trap-assisted recombination dynamics after the push. However, as discussed in response to the previous points (i and ii), the sub-ps cooling dynamics appear to be less sensitive to the passivation as compared to the recombination dynamics. The overall process could be described by our model, where the hot

carriers (HCs) are trapped during the cooling and how such process leads to faster recombination for the cold carriers, without any change to the cooling times. We can speculate that the trapped state (*i.e.*, the trap potential energy surface) displays a similar electron-phonon coupling strength, thus leading to similar cooling times on the trap potential energy surface. On the other hand, the trapped-assisted recombination process is governed by Coulomb interactions, which are expected to increase after trapping.

Based on this feedback, we have made the following revisions into the manuscript:

- Page S13: **Figure R.3.1** was added as Figure S16b
- Page 18, Line 368: Further discussion on the comparison between PPP data for pristine and ligand exchange MAPbI₃ NCs is added "*Hot carrier-induced trapping processes provide an additional and alternative cooling pathway that intuitively should speed up overall hot carrier cooling times. However, direct comparison between PPP data for pristine and ligand-exchange MAPbI₃ NCs (Figure S16b) indicates that while the long-time recombination dynamics are affected by the passivation, the cooling dynamics are unchanged within our time resolution. Similar results for the cooling were recently reported by Harel et al. for bulk MAPI.¹⁹ Hence, we speculate that defects passivated by TOPO do not significantly affect the electron-phonon coupling, and therefore do not significantly speed up the cooling times.*"

2. I see the authors have recently published work using the same pump-push-probe experiment on bulk MAPbI₃ (Ref. 40 in the manuscript). Out of curiosity, how do the push-induced TA signals for bulk MAPbI₃ compare with the analogous NCs? This comparison may also help to shed light on the impact of the NC surface on cooling.

We thank the Reviewer for the interest in the work from our lab, and we agree that the proposed comparison is indeed interesting. The mentioned study (Ref. 40) focused on the interface energy barrier for HC extraction from bulk MAPbI₃ film into a Bphen layer and did not directly address the long-time dynamics. Although it was not discussed in the paper, we observed signatures of faster carrier recombination after the introduction of push pulse. Additionally, these enhanced recombination rates were found to be dependent on sample qualities. Such observation implies that a similar trapping mechanism could also be involved in MAPI bulk films, where traps commonly originate from grain-boundaries and surfaces.

However, we also noticed an essential difference in the sub-ps push cooling dynamics for the visible (2.07 eV) push excitations. In the bulk MAPI (Ref. 40, Fig. 2a), we observed a partial sub-ps recovery of the push-induced signal, while we did not observe a similar process for MAPbI₃ NCs. We ascribe this different photophysics to the peculiar surface of NCs, where ligand and other types of defects can assist processes similar to the photoionization (*i.e.*, hot carriers removal from the NCs). This is beyond the scope of the current study, and we can anticipate that further studies are underway to present a detailed comparison between quantum confined and bulk systems.

3. Please label the data in all the figures as "MAPbI3 NCs" or "MAPbBr3 NCs" (not just MAPI and MAPB) to avoid confusion with the bulk materials.

We thank the Reviewer for the pointer. We harmonized the labeling throughout the manuscript and SI.

4. If the results of this work are to be believed, it appears that the consequences of hot carrier trapping are mostly manifested at "late" times, i.e. after cooling has occurred. In a hot-

carrier solar cell, extraction should outpace cooling. Could the authors provide some discussion connecting their results with this practical consideration?

In a nutshell, our findings show that it is easier for energetic/hot carriers to be trapped in perovskite NCs. Moreover, the similar cooling dynamics (within our time resolution) from PPP between the pristine and ligand-exchanged samples indicate that the consequences of this trapping mostly manifest at longer time delays. Overall, this results in reduced efficiencies for applications that rely on the broadband absorber nature of PNCs and usually assume that no carriers are lost during the cooling.

The Reviewer raises a different point regarding possible implications of our spectroscopic results on devices like HC solar cells. We believe that the presence of hot carrier traps could potentially hamper the development of HC technologies. While cooling dynamics do not seem to be strongly affected, for the reasons explained above, the presence of an additional relaxation pathway could still impact the hot carrier temperatures, which cannot be studied using PPP. For instance, a very recent report (*Nano Lett.* 2019, 19, 2, 684-691) suggests that, in perovskite solar cells, passivation can increase hot carrier temperatures, while leaving cooling times unaffected. As shown in **Figure R.3.2**, the additional passivation provided by TOPO to MAPbI₃ NCs causes an increase in the HC temperatures with respect to the pristine MAPbI₃ NCs in the same experimental conditions.

Figure R.3.2 Comparison between hot carriers temperatures for pristine MAPbI₃ NCs (blue) and TOPO-ligand exchanged MAPbI₃ NCs (red). TA measurements were obtained using 3.1 eV (80 μJ/cm²) and white light probe. HC temperatures were extracted using the Boltzmann fit method.

Hence, these preliminary results suggest that the presence of HC losses could impact HC solar cell technologies. We believe that this is an interesting scenario, and we envisage further systematic studies on this matter. However, this falls outside of the scope of the present work.

Based on this feedback, we modified the discussion:

- Page S13: We added **Figure R.3.2** in the SI as Figure S16e
- Page 18, Line 370: *Nevertheless, proper passivation is also crucial for HC solar cells, ASE and lasing applications. Indeed, as suggested by the passivation effect of HC temperatures (Fig. S16e), HC carrier losses could still affect these applications in terms of their efficiencies (i.e., reducing the number of carriers available for extraction or photon conversion, respectively).⁴ Future efforts in studying the passivation effects on these applications are needed.*

5. Can the authors comment on why, in Figure S11(a), the PPP trace goes below zero norm $\Delta T/T$ at 3 ps?

The Reviewer correctly observes that the PPP trace peaks at negative values during the push re-excitation. For MAPbBr₃ NCs, the TA signal close to the main photobleaching (PB) peak (probe energy 2.3 eV) is complex and comprises overlapping spectroscopic signals. The presence of a strong and short-lived photoinduced absorption (PIA) signal in that spectral region upon non-resonant excitation, is widely reported in the literature and usually ascribed to the presence of hot carriers (*J. Am. Chem. Soc.* 2019, 141, 3532–3540). To confirm this, we report the complete trace comprising the early-time dynamics, which clearly shows the presence of this short-lived PIA (**Figure R.3.3**). Hence, the presence of a negative PPP trace can be explained as the effect of carrier re-heating, upon push absorption. The presence of hot carriers in the systems causes that signal to revert to PIA, by reducing the PB signal. Unfortunately, given the differential shape of the PPP delta signal, meaningful PPP kinetics are only found where the main PB signal overlaps with this early time PIA.

Figure R.3.3: Normalized pump-push-probe kinetics for MAPbBr₃ NCs in anhydrous toluene solutions. The sample was pumped at 3.10 eV (40 $\mu\text{J cm}^{-2}$), pushed at 1.03 eV (2.0 mJ cm^{-2}), and probed at 2.3 eV. The early time dynamics reveal the presence of a PIA, associated with the presence of hot carriers.

Based on this feedback, we have made the following revisions into the manuscript:

Page S9: **Figure R.3.3** was added as Figure S11b

6. Reviewer #2 raised an interesting point about the Stark effect. The authors stated in their response that this would lead to a "differential-like transient signal around the band-edge of the material", but claimed that it is not significant in the presented data. How do the authors explain the trend in Figure S14(b), which shows a clear derivative-like signal, redshifted with respect to the bandgap of the MAPbBr₃ NCs? Previous pump-push-probe studies by the Friend group on organic semiconductors have attributed these effects to Stark shifting.

The Reviewer raises an interesting point regarding the nature of the signal observed in Figure S14b. Before commenting on this point, we would like to stress that it is clearly stated in the main text the use of PPP on MAPbBr₃ NCs presents some limitations, and therefore we focused on MAPbI₃ NCs as a model system.

Having this said, we partially agree with the Reviewer: there could be a contribution from the optical Stark effect (OSE) to the observed signal, especially due to the presence of strongly bound excitons, and thus of a strong-light matter coupling, in our MAPbBr₃ NCs samples.

More in detail, the observed derivative-like signal represents a blueshift of the excitonic transition induced by the push. (More clear in Figure S12b) However, it should also be considered that the detuning energy for this experiment is considerably large (i.e., ~1.3 eV) and that the magnitude of the shift is inversely proportional to the detuning energy (Sci. Adv. 2016;2:e160047). A conservative estimate of the Rabi energy at the presented push fluence is ~63 meV [estimated from 2D (C₆H₅C₂H₄NH₃)₂PbBr₄ perovskites coefficients, which have the highest reported oscillator strengths in the perovskite family]. From this upper limit value,

we can estimate a maximum blue shift of ~ 2 meV by OSE at this fluence and with this detuning energy. However, from our pump-push-probe data, we observe experimentally a ~ 10 meV blueshift (Figure S12b), which suggests that OSE cannot solely account for the observed response.

We propose the transient reduction of bandgap renormalization caused by push-induced heating of the excitons as a possible contributing effect. Indeed, the presence of push-induced hot carriers/hot-excitons is expected to reduce the bandgap renormalization effect due to reduced electron-hole Coulombic interaction, which is strong in excitonic systems. (DOI: 10.1006/spmi.1996.0052) This assignment is consistent with our separate finding (accepted and to be published soon in *The J. Phys. Chem. Lett.*), where we observed an increase and saturation of bandgap-renormalization concomitant with the cooling process.

This hypothesis is further confirmed by the observed cooling/recovery time of the push-induced signal $\sim 300 \pm 90$ ps in our samples, which is beyond the pulse duration (< 90 fs). The longer time duration of the optical response rules out its fully non-resonant nature. Namely, if the signal was due to the sole OSE, we would expect a signal coinciding with the temporal pulse duration, as the OSE would only occur within the pulse. However, this is not the case –

Figure R.3.4.

Figure R.3.4: (b) Exponential fit of the differential pump-push-probe kinetics for MAPbBr₃ NCs in anhydrous toluene solutions. The sample was pumped at 3.10 eV (40 μJ cm⁻²), pushed at 1.03 eV (2.0 mJ cm⁻²), and probed at 2.45 eV. The pink area represents the deconvolved instrument response function from the fit.

As a final note, the Reviewer refers to papers from Richard Friend's group, which suggest the role of the Stark shift. In these papers [e.g., Nature Materials volume 16, pages 551–557(2017)], authors investigate the quadratic Stark effect (QSE). Although QSE is similar to OSE, the former originates from carrier-related electric fields rather than from optical fields. Namely, in these studies, the push excites the carriers, transferring them energy, and changing their spatial and energetic distribution. This excitation perturbs the internal electric field, thereby resulting in an electroabsorption-like derivative signal. These reports further substantiate our interpretation, which relies on the push-induced modification of bandgap renormalization. Specifically, in our case, the push re-heating of excitons in MAPbBr₃ NCs reduces Coulombic correlations in these systems. This explains the transient blueshift in terms of reduces QSE (i.e., reduced bandgap renormalization).

Therefore, although we cannot 100% rule out the presence of OSE, we believe that the signal is due to the interplay of more complex population dynamics. In conclusion, we can agree with the Reviewer in saying that the nature of the PPP response of MAPbBr₃ NCs is complex and not easily rationalized. However, as mentioned previously, this does not affect our conclusions.

On a non-scientific note, I appreciate that the authors may not be native speakers, but I still have gripes with the poor grammar in the manuscript. I have identified the following recurring issues, which I hope the authors will find constructive for any future iterations of their paper: i) General sentence structure. Example: on line 230, "the presence of

overlapping spectroscopic responses makes challenging their complete disentanglement..." should read "the presence of overlapping spectroscopic responses makes their complete disentanglement challenging...")

ii) Tense of clauses. In particular, the mixing of past and present tense within the same sentence. Example: on line 13, "Amongst the many spectacular properties of the hybrid lead halide perovskites, their defect tolerance is revered as the key enabler for a spectrum of high-performance optoelectronic devices" is in the present tense, but the remainder of the sentence on line 15 "that propelled perovskites to stardom" is in the past tense.

iii) Incorrect use of singular and plural terms. Example: on line 62, "Perovskite nanocrystals (PNCs) provides an exciting platform..." should read "Perovskite nanocrystals (PNCs) provide an exciting platform..."

In my opinion, the current manuscript is difficult to read, and the basic language issues above need to be addressed to meet the standards of Nature Communications.

We thank the Reviewer for spotting the typos which unfortunately had slipped through us (again) and we are saddened that the abovementioned grammar errors had made him/her uncomfortable. After double-checking the grammar, we believe that language issues have now been resolved and that our paper meets the standards of *Nature Communications*.

REVIEWERS' COMMENTS:

Reviewer #3 (Remarks to the Author):

Thank you to the authors for responding to my queries. In my view, there are still some minor grammatical issues, but I will not stand in the way and give the manuscript my blessing for publication in Nature Communications.

The authors may be interested to compare and contrast their findings with a very recent publication on pump-push-probe in perovskite nanocrystals from Bakulin's group: [10.1021/acs.nanolett.9b04491](https://doi.org/10.1021/acs.nanolett.9b04491). This paper also indicates that the surface states do not play a major role in intraband cooling.

Reviewer #3

Thank you to the authors for responding to my queries. In my view, there are still some minor grammatical issues, but I will not stand in the way and give the manuscript my blessing for publication in Nature Communications.

We are delighted by Reviewer #3's encouraging comments and constructive scientific feedback. We are greatly appreciative for all his/her inputs that helped to raise the standards of our manuscript.

The authors may be interested to compare and contrast their findings with a very recent publication on pump-push-probe in perovskite nanocrystals from Bakulin's group: [10.1021/acs.nanolett.9b04491](https://doi.org/10.1021/acs.nanolett.9b04491). This paper also indicates that the surface states do not play a major role in intraband cooling.

We thank the Reviewer for the suggestion. The very recent paper by Hopper et al. is indeed interesting and deserves a mention in our manuscript. Authors report a pump-push-probe (PPP) study on hot carriers cooling times in FA and CsPbX₃ NCs. Among other observations (the paper mostly concerns the effect of hot and cold carriers on the cooling times), they also observe that passivation does not significantly affect the cooling times, thereby validating what we report in our Supplementary Information. However, as discussed in the previous revision round, similar cooling times do not rule out the presence of hot carrier trapping, which can result in lower carrier temperatures (that cannot be extracted by PPP kinetics, as the one reported in the mentioned paper) and in enhanced recombination dynamics. Unfortunately, the PPP experiments reported in that paper do not extend to longer time delays and therefore no further comparison can be driven.

Once again, we would like to thank Reviewer #3 for all the valuable feedback provided in these revisions.